

# A multimillennial Alpine ice core chronology synchronized with an accurately dated Arctic Pb record

Paolo Gabrielli[1,*], Theo M. Jenk[2,3,*], Michele Bertó[4], Giuliano Dreossi[4], Daniela Festi[5], Werner Kofler[5],

Mai Winstrup[6], Klaus Oeggl[5], Margit Schwikowski[2,3,7], Barbara Stenni[4] and Carlo Barbante[4,8]

[1]Italian Glaciological Committee c/o University of Turin, Turin, Italy

[2]Center for Energy and Environmental Sciences, Paul Scherrer Institut, 5232 Villigen PSI, Switzerland

[3]Oeschger Centre for Climate Change Research, University of Bern, 3012 Bern, Switzerland

[4]Department of Environmental Sciences, Informatics and Statistics, Ca' Foscari University of Venice, Venice-Mestre, 30170, Italy

[5]Institute for Botany, University of Innsbruck, Innsbruck, 6020, Austria

[6]DTU Space, Technical University of Denmark, Kongens Lyngby, 2800, Denmark

[7]Department of Chemistry and Biochemistry and Pharmaceutical Sciences, University of Bern, 3012 Bern, Switzerland

[8]Institute of Polar Sciences-CNR, Venice-Mestre, 30170, Italy

*Equally contributed to this manuscript

*Correspondence to:* Paolo Gabrielli (paologabrielli@hotmail.com), Theo Jenk (theo.jenk@psi.ch) and Carlo Barbante (barbante@unive.it)

**Abstract.** A low latitude-high altitude Alpine ice core record was obtained in 2011 from the glacier Alto dell'Ortles (3859 m, Eastern Alps, Italy). A preliminary absolute timescale (TC2016) based on a peak in $^3$H activity, and $^{210}$Pb and $^{14}$C dating of carbonaceous particles and organic remains provided evidence of one of the oldest Alpine ice core records spanning the last ~7000 years, back to the last Northern Hemisphere Climatic Optimum. Here we provide an additional number of time markers that corroborate the multimillennial nature of the Alto dell'Ortles ice cores while significantly decreasing the uncertainty of the chronology. First, $^{14}$C dating of an additional organic fragment (a charred spruce needle) discovered next to the basal ice provided an age (232 ± 126 BCE) which agrees with previous $^{14}$C dates in the oldest part of the record. Second, novel seasonally resolved pollen records from the upper firn/ice portion of the Alto dell'Ortles cores were combined with $\delta^{18}$O and dust annual variations to refine the dating for the 20$^{th}$ century by means of an automatic algorithm (Straticounter; between 1927 and 2011 CE) and visual counting (from 1900 to 1926 CE). The new and previous time markers were combined into a revised





intermediate timescale (CP2025/1) by fitting using Markov chain Monte Carlo simulation (COPRA model). CP2025/1 was used for synchronizing a novel Pb concentration record obtained from the Alto dell'Ortles cores with a Pb record from an array of Arctic ice cores (AN), well-dated (±5 years) for the ~200 BCE to ~1900 CE period. The ties used for matching the two Pb records were within the uncertainty of CP2025/1 and of the selected tie-points (1- to 2-sigma, in the ancient part; 1-sigma, in

the recent part). The correlation obtained after synchronization is 0.44 (Pearson's r, p < 0.001), demonstrating that these two distant atmospheric Pb records share a large portion of their variability back to 200 BCE. The synchronization of CP2025/1 with AN resulted in a, further refined, final timescale (CP2025/2). An investigation of CP2025/1 and CP2025/2 by means of a simple 1-D flow model suggests that non-steady-state conditions, in particular changes in past net accumulation rates, need to be considered to provide a full physical explanation of the age-depth relationship obtained. The new Alto dell'Ortles

CP2025/2 chronology of improved accuracy will allow to constrain Holocene climatic and environmental histories emerging from this high-altitude glacial archive of Central Europe. The novel combination of methodologies used may also be adopted to build, or improve, the chronologies of other ice cores extracted from-low latitude/high-altitude glaciers that typically suffer from larger dating uncertainties compared with well dated polar records.

## 1 Introduction

Ice cores extracted from polar regions and high-altitude/low-latitudes glaciers are archives of past climatic-environmental histories as they record physical, chemical and biological characteristics of the past atmosphere. To interpretate this information, it is of fundamental importance to link the englacial depth of the various ice sections to the timing of the original snow deposition. This provides a function that is commonly known as chronology.

Counting annual layers, establishing time markers (e.g. volcanic horizons, $^{10}$Be, $^{3}$H peaks etc.), synchronizing with other dated paleo-records, and developing ice flow models are widely used methods to precisely and accurately date ice cores from polar regions (Parrenin et al., 2007; Svensson et al., 2020), where the negligible/slow horizontal flow at the drilling sites and the largely below freezing englacial temperatures allow a full physical-chemical preservation of the ice stratigraphy and the time markers embedded within the accumulated snow layers. In contrast, dating ice layers from high-altitude/low-latitude

glaciers is more challenging because of: i) their smaller ice thicknesses, implying that counting annual layers is usually limited to the upper 50 to 100 meters (typically covering only a few centuries) (Schwikowski et al., 2014); ii) a larger horizontal flow at the drilling sites that can quickly alter/disrupt the original ice stratigraphy (Thompson et al., 2000), and; iii) increasingly widespread post depositional processes that are linked to modern warmer air temperature and summer meltwater percolation which can overprint (or prevent the full conservation of) the time markers embedded in the glaciers (Gabrielli et al., 2010).


In 2011 we extracted four 60-75 meter long ice cores located within 10 m from each other from the Alto dell'Ortles glacier (3859 m), near the summit of Mt. Ortles (3905 m) in the Eastern Alps, Italy. By using the three ~75 m longest cores (from now on cores #1, #2 and #3; with core #4 of only ~60 m being archived for future analyses) we demonstrated that the upper firn portion of the Alto dell'Ortles glacier was temperate, with intense summer meltwater percolation that possibly affected this drilling site since the 1980s. However, the underlying ice was cold (-2.8 °C near the bedrock) (Gabrielli et al.,



2012) which preserved stratigraphy that was thousands of years old (Gabrielli et al., 2016). Our discovery of millennial-age ice in the Eastern Alps was confirmed by another ice core record extracted from the nearby drilling site of Weißseespitze (3500 m, Austria) (Bohleber et al., 2020).

In order to develop an initial chronology, we previously identified well-defined $^3$H and beta activity peaks at 41 m
depth from atmospheric thermonuclear testing attributed to 1963 and used the activity of $^{210}$Pb to date the Alto dell'Ortles cores to 59 m depth (~1930 CE) (Gabrielli et al., 2016). In addition, determination of a $^{14}$C age of a larch needle embedded near the basal ice, along with $^{14}$C dating of several ice samples (using the water insoluble organic carbon fraction entrapped; WIOC-14; Uglietti et al., 2016; Jenk et al., 2009; Hoffmann et al., 2018) allowed us to obtain absolute time markers from the deepest ice, the age of which was calculated to date back to ~7000 years. All the age constraints obtained were used to build
an initial chronology (TC2016 from now on) by means of Markov chain Monte Carlo simulation (COPRA model; Breitenbach et al., 2012), which also provided time uncertainty as a function of depth from a few years in the upper modern firn layers to ± 500 years in the deepest portion of this ice core (Gabrielli et al., 2016).

A large uncertainty in TC2016 occurred not only in the basal layers of the ice core but also in the intermediate depths (60-72 m), dated between 1900 and 200 CE (e.g., ~ ± 50 years at 1850 CE, ~ ± 100 at 1650 CE and ~ ± 400 at 500 CE). The
relative time uncertainty (50-70%) of the intermediate portion of the record was higher than both the modern (1900 -2011 CE, 5-25% uncertainty) and the oldest (200 CE – 5000 BCE; 10-30% uncertainty) sections. Altogether, this large absolute and relative uncertainty posed questions on the ability of TC2016 to reconstruct precise climatic and environmental histories of the Alps and Central Europe from these cores. In particular, the uncertainty in the intermediate portion of the cores is essentially due to i) the lack of time markers such as volcanic horizons; and ii) the use of only two WIOC-14 $^{14}$C ages (1355 ± 205 CE
and 429 ± 286 CE) above 72 m depth (after 200 CE). These ages were also characterized by relatively large uncertainty due to the intrinsic limits of the $^{14}$C technique to date organics that were centuries to millennia old (Uglietti et al., 2016).

Here we report additional stratigraphic/time markers, the use of which confirms TC2016 within the time uncertainty presented. Using the same approach, an updated version of TC2016 was first build as an intermediate timescale, in the following named CP2025/1, gaining from an improved depth alignment of the three Alto dell'Ortles cores used, for which a
greatly increased number of tie points obtained from the δ$^{18}$O records was utilized, and newly available time markers. In particular, the new age constraints were obtained from: 1) $^{14}$C dating of an additional organic fragment (a charred spruce needle; 232 ± 126 BCE) discovered near the bottom ice (72.82 m) in core #1; and 2) the automated counting of annual layers with Straticounter (Winstrup et al., 2012; Winstrup, 2016) based on pollen, δ$^{18}$O and dust records from core #1 from 2011 to 1927 CE (51 m depth in core #1) and visual counting from 1926 to 1900 CE (57 m depth in core #1). We used CP2025/1 for
synchronizing the high resolution crustal excess Pb concentration record in core #3 (which was substantiated by a second, less resolved Pb record from core #1) to a corresponding well-dated (±5 years) Arctic record from ~1900 CE to ~200 BCE (McConnell et al., 2019). This allowed us to obtain a further refined final Alto dell'Ortles time scale (CP2025/2). Finally, we applied the Dansgaard-Johnsen glaciological ice flow model (Dansgaard and Johnsen, 1969) for the Alto dell'Ortles drilling site to investigate the physical nature of the revised empirical chronologies CP2025/1 and CP2025/2.



## 2 Alto dell'Ortles ice cores re-alignment

The first step to develop a more accurate chronology was to revise the previously published (Gabrielli et al., 2016) common depth scale for the three Alto dell'Ortles cores, using core #2 for depth reference as we did for TC2016 (see Supplementary Text 1). Depth realignments of core #2 with cores #1 and #3 were performed in two steps using the Analyseries 2.0.8 software (Paillard et al., 1996). First, the $\delta^{18}O$ records were more finely matched using additional tie points between cores #1 and #2 (for a total of 122 points) and between cores #3 and #2 (87 points). In this case the 122 tie points between cores #2 and #1 provide a Pearson correlation r = 0.78 (significant at p < 0.01) between all the $\delta^{18}O$ values aligned on core #2 depth while 87 tie points between cores #2 and #3 give r = 0.79 (p < 0.01). For comparison, in TC2016 the 17 tie points between cores #2 and #1 provided r = 0.72 (p < 0.01) while 14 tie points in cores #2 and #3 gave r = 0.67 (p < 0.01). Overall, the increase in r for CP2023 suggests a refinement of the depth alignments compared to the one performed previously for TC2016 as illustrated in Fig 1.

While a more detailed alignment was obtained (Fig. 1, blue vertical lines), a lack of tie points persisted within the 60-73 m depth interval between cores #2 and #3. In TC2016 this was a problem because, as we realized later, this resulted in a depth misalignment of a few tens of cm, which resulted in a significant time lag (up to ~300 years) between the three cores (see Supplementary Text 1). To better align the $\delta^{18}O$ #2 and #3 records in this interval, 31 additional tie points were obtained (Fig. 1, red vertical lines) from two high resolution Pb concentration records independently determined in cores #1 and #3 by discrete Inductively Coupled Plasma Mass Spectrometry (ICP-MS) and continuous flow analysis (CFA) ICP-Sector Field MS (ICP-SFMS) at the University of Venice and at the Ohio State University, respectively (Gabrieli, 2008; Gabrieli and Barbante, 2014) (Fig. 2). Pb was not determined in core #2 due to insufficient ice volume. The environmental interpretation of the Pb record is not within the scope of this paper and will be discussed in detail in a separate manuscript.

We note that the observed differences in Pb concentrations between cores #1 and #3 are probably due to the different acid leaching time between continuous flow analysis (CFA; using online acidification; core #3) and discrete analysis (adopting pre-acidification of aliquots during sample preparation; core #1). In any case, different acidification methods do not affect Pb trends and features (maxima, minima, Pb variations) used for wiggle matching. The different spatial resolution obtained by using either Pb determination method (~ 0.2 cm and ~ 4 cm in the cores #3 and #1, respectively) potentially causes still slight depth misalignments which are nevertheless negligible with respect to the final temporal uncertainty within this depth interval (a few years compared to around 10% of the age at this depth, see below). Thus depth alignment between 60 and 73 m was performed in three steps: i) by matching the Pb concentration records in cores #1 and #3 (31 tie points obtained) using Analyseries 2.0.8 software (Paillard et al., 1996) (Fig. 2); ii) by transferring the depths of the 31 Pb points in core #1 to core #2 using the depth map obtained by matching cores #1 and #2 using $\delta^{18}O$, see above; and iii) by linking the $\delta^{18}O$ stable isotope records in cores #2 and #3 over this interval using as a guide the depth



of the 31 supplemental Pb tie points. The final correlation obtained within this depth interval between the two Pb records is r = 0.91 (p < 0.01).

In conclusion, we note that the significant correlations between the δ¹⁸O and Pb records from Alto dell'Ortles cores provide further evidence of their high degree of reproducibility. This reproducibility also confirms the lack of stratigraphic disturbance due to subsurface ice flow at this drilling site (Gabrielli et al., 2016).

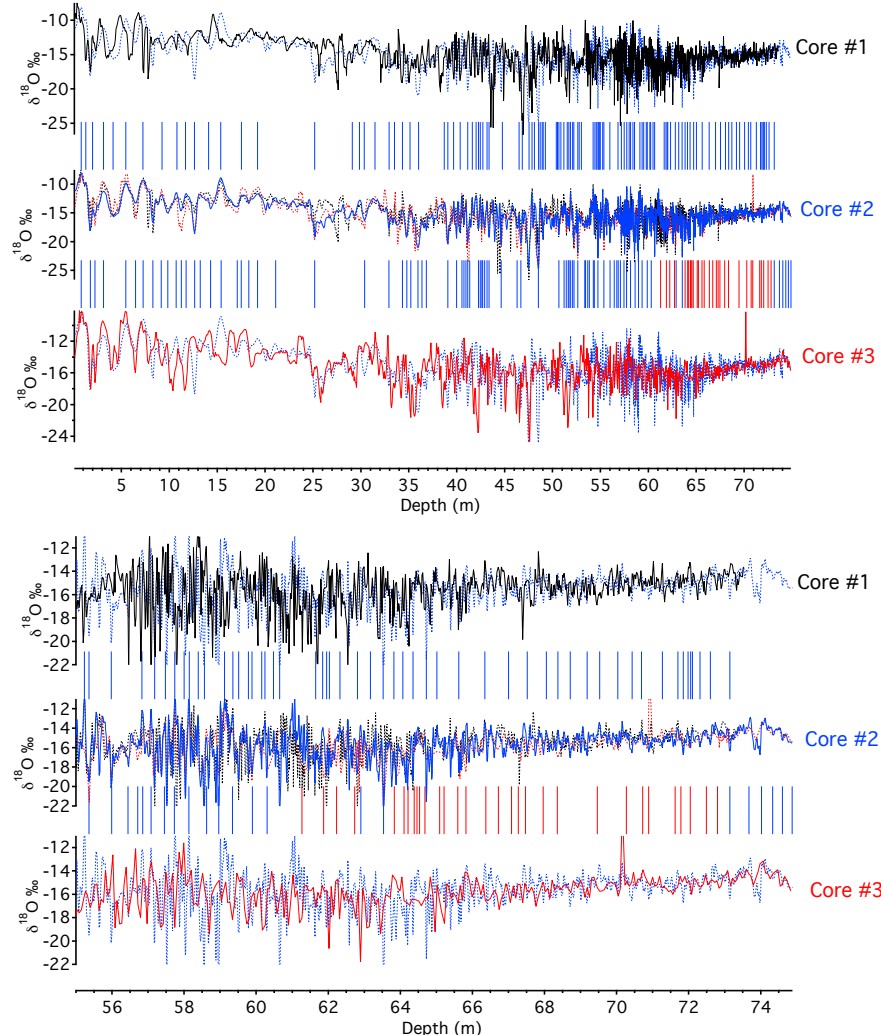

Figure 1: Revised depth alignments using the δ¹⁸O records from the Alto dell'Ortles cores #1 (in black), #2 (in blue) and #3 (in red) over the entire lengths (upper panel) and the bottom portions (lower panel) of the cores. A common core #2 depth (X axis) is used in the mid panel displaying the three matched records while the δ¹⁸O records in the top and bottom panels are shown on their original depth scales. Blue vertical lines indicate the δ¹⁸O tie points while red lines indicate tie points obtained through the Pb records of cores #1 and #3 (see Fig. 2). Adapted from Fig. 10 in Gabrielli et al., (2016).





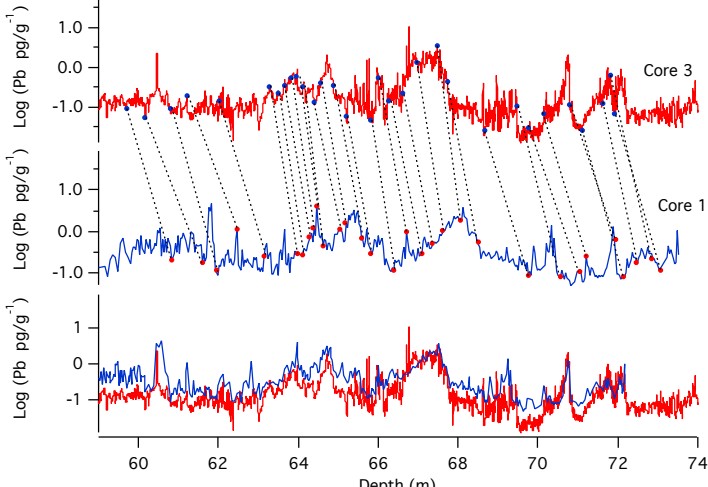

Figure 2: Pb concentrations in the deep portion of cores #3 (red; Continuous Flow Analysis ICP-SFMS, the Ohio State University) and #1 (blue; discrete ICP-MS analysis, University of Venice) at their respective #1 and #3 depth scales (upper two panels) and matched to core #3 depth (lower panel).

## 3 Time markers

After the improvement of the depth alignment of the three Alto dell'Ortles cores, we combined the original time markers from TC2016 and new markers (Table 1) to construct CP2025/1. In the deepest portion a new [14]C age (232 ± 126 BCE) was included (see paragraph 3.1; Table 2) and to refine the chronology from 1900 to 2011 CE, we retained the beta and

10 [3]H activity peaks of 1955 and 1963 CE from TC2016, added three additional time constraints (2006, 1995, 1986 CE; Table 3) and 113 new annual time markers obtained by instrumental and visual annual layer counting (ALC) of pollen, dust and $\delta^{18}$O (Paragraph 3.2; Supplementary Tables 1-2).

| Parameter | Use | Core #1 (73.53 m) | Core #2 (74.88 m) | Core #3 (74.83 m) |
|---|---|:---:|:---:|:---:|
| Stable isotopes | Depth aligment; annual layers | ✓ | ✓ | ✓ |
| Pollen concentration | Annual layers; events | ✓ | | |
| Pollen day of the year, 32 types (DOY 32) | Annual layers | ✓ | | |
| Pollen Day of the Year, 46 types (DOY 46) | Annual layers | ✓ | | |
| Pollen depth-to-day matching (DOY match) | Annual layers | ✓ | | |
| Dust | Annual layers | ✓ | | |
| [210]Pb | 20[th] century dating | | ✓ | |
| [3]H | Event | ✓ | ✓ | |
| Beta activity | Events | | ✓ | ✓ |
| Pb | Depth alignment | ✓ | | ✓ |
| [14]C in water insoluble C organic fraction (WIOC-14) | Preindustrial dating | ✓ | | ✓ |
| [14]C in organic macrofragments | Preindustrial dating | ✓ | | |

15 Table 1: Empirical parameters used to date the Alto dell'Ortles ice cores.




3.1 The new [14]C time marker near the bottom ice (232 BCE)

An intact larch needle at 73.25 m in core #1 provided a [14]C calibrated age of 659 ± 102 BCE (Gabrielli et al., 2016), which was useful for dating the older part of TC2016. More recently we extracted a charred spruce needle from core #1 at 72.82 m (Fig. 3). Spruce are very common conifers in the Alps, including the Mt. Ortles area. Atmospheric vertical convection

most likely transported the two needles from a lower elevation to the Alto dell'Ortles glacier. The charring of the needle likely resulted from a local forest fire, the heat from which may have produced or intensified atmospheric convection. This fragment, which contained 57 µg of carbon, was radiocarbon dated at the Paul Scherrer Institute using the Bern AMS facility (LARA Laboratory, University of Bern, Switzerland) (Szidat et al., 2014) and provided a conventional [14]C calibrated age of 232 ± 126 BCE (Bern AMS sample number BE-12451.1.1). This date is stratigraphically consistent with the [14]C date (659 ± 102 BCE)

of the first larch needle found at 73.25 m in core #1 and with the shallower/younger and deeper/older WIOC-14 ages (see Table 2). Within the range of uncertainty, the [14]C calibrated age (232 ± 126 BCE) of the charred needle found at 72.82 m is consistent with the TC2016 dating (146 ± 370 BCE) at the same depth. Thus, these new [14]C data fit well within the existing sequence, confirming the continuous increase of age with depth that was observed previously (Gabrielli et al., 2016). In addition, we can exclude the possibility of stratigraphic disruption due to ice flow at this depth, at least within the [14]C age

uncertainty. All [14]C ages were calibrated using the OxCal program v4.3.2 (Bronk Ramsey, 2009) with the IntCal20 radiocarbon calibration curve (Reimer et al., 2020) and reported uncertainties indicate the 1σ range.

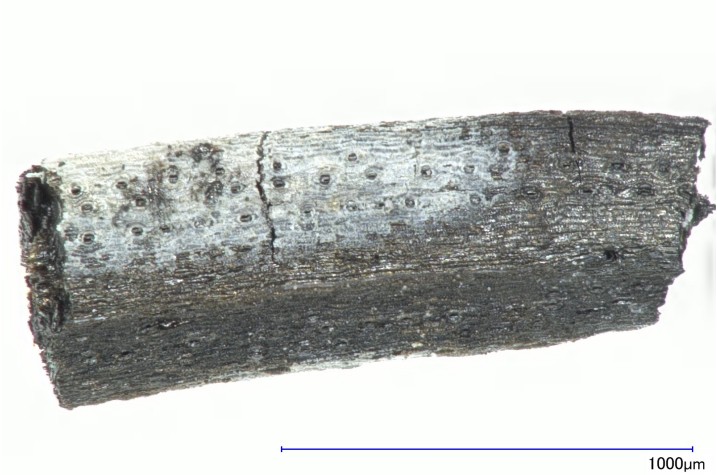

1000µm

Figure 3: The charred spruce needle fragment found in core #1 at 72.82 m depth and [14]C dated to 232 ± 126 (1σ) BCE. The image was produced by using a Kayence VHX 2000 digital microscope at the University of Innsbruck.



| Core # | Tube # | Measure | Top depth (m) | Bottom depth (m) | Ice mass (kg) | C mass* (µg) | F¹⁴C | ¹⁴C age (yrs BP) | Cal age** (yrs cal BP)- 1σ range | µ cal age (yrs cal BP) | µ cal age (yrs cal b2012) | µ cal age (CE-BCE) | σ (yrs) | TC2016 | CP2023 |
|---|---|---|---|---|---|---|---|---|---|---|---|---|---|---|---|
| 1 | *98b* | WIOC-14*** | 68.26 | 68.96 | 0.31-0.34 | 15.1-17.9 | *0.927 ± 0.025* | *609 ± 217* | *(330 - 784)* | *589* | 651 | 1361 | *204* | ✓ | ✓ |
| *3* | *102* | *WIOC-14**** | 70.87 | 71.57 | 0.26-0.28 | 7.2-13.3 | *0.823 ± 0.027* | *1565 ± 264* | *(1178 - 1781)* | *1508* | 1570 | 442 | *288* | ✓ | ✓ |
| 1 | *103b* | WIOC-14**** | 71.77 | 72.48 | 0.93 | 10.37 | 0.932 ± 0.037 | 569 ± 320 | (155 - 903) | 566 | 628 | 1384 | 289 | | |
| 1 | *104b* | Charred spruce needle | 72.78 | 72.82 | - | 57 | 0.761 ± 0.009 | 2193 ± 98 | (2333-2070) | 2182 | 2244 | -232 | 126 | | ✓ |
| 1 | *105b* | Larch needle | 73.25 | 73.25 | - | 68 | 0.728 ± 0.006 | 2550 ± 65 | (2499 - 2751) | 2609 | 2671 | -659 | 102 | ✓ | ✓ |
| 3 | 106 | WIOC-14 | 73.73 | 74.02 | 0.30 | 10.91 | 0.628 ± 0.031 | 3737 ± 397 | (3578 - 4786) | 4171 | 4233 | -2221 | 524 | ✓ | ✓ |
| 3 | 106 | WIOC-14 | 74.02 | 74.24 | 0.29 | 11.50 | 0.568 ± 0.030 | 4544 ± 424 | (4620 - 5718) | 5176 | 5238 | -3226 | 531 | ✓ | ✓ |
| 3 | 106 | WIOC-14 | 74.24 | 74.47 | 0.31 | 18.47 | 0.481 ± 0.020 | 5879 ± 334 | (6321 - 7156) | 6739 | 6801 | -4789 | 364 | ✓ | ✓ |

\* Pure C extracted and available for AMS analysis after combustion.

\*\* ¹⁴C calibration was performed in OxCal v4.4.4 (C.B. Ramsey, 2021) with IntCal20, the Northern Hemisphere atmospheric calibration curve (Reimer et al., 2020).

\*\*\* Provided ice sample and C mass indicates the range for the three subsamples of 98b and 102, and F14C (and resulting ages) indicates their combined result (Gabrielli et al., 2016; Uglietti et al., 2016).

\*\*\*\* Outlier (values in brackets), explained by an exceptionally large ice sample volume processed, and a resulting small C mass available for a single AMS measurement (Gabrielli et al., 2016).

Table 2: Information on ¹⁴C analyses of organic material from the Alto dell'Ortles cores (adapted from Table 2 in Gabrielli et al., 2016). Tubes #104b and #105b in core #1 refer to the respective sections in which the two conifer needles were found. All other ¹⁴C ages were obtained from analysis of the water insoluble organic carbon fraction (WIOC-14) extracted from the corresponding ice sections of cores #1 and #3. All radiocarbon dates were (re)calibrated using the IntCal20 calibration curve (OxCal v4.3.2). The reported uncertainties (i.e. the age range for calibrated age) indicate the 1σ range with µ-age denoting the age of highest probability density. For the selection of data points used for the revised timescale refer to text and to Gabrielli et al., (2016).

### 3.2 Annual layer counting (1900-2011 CE)

A clear seasonal pollen signal was reported from the shallow firn temperate layers (10 m depth) in the Alto dell'Ortles glacier (Festi et al., 2015; Festi et al., 2017). New pollen records of relative and total concentration values were obtained from core #1. Despite recent intense summer meltwater percolation, the seasonality of the pollen record appears to be conserved in the firn and in the lower ice portion. This is demonstrated by pollen concentration, $\delta^{18}O$ and dust concentration identified within the well-dated 1955-1963 CE interval which was constrained by beta and ³H activity peaks. These three parameters consistently show high values during the warm seasons (late spring to late summer) and low values during the cold seasons (Fig. 4). In particular, pollen age-markers correspond to winter/spring of each given year as pollen concentration in the atmosphere is high during spring/summer, while little or no pollen is present in winter. This situation is reflected in the corresponding seasonal ice core layers.

We applied the StratiCounter algorithm (Winstrup et al., 2012; Winstrup, 2016) to a combination of pollen concentrations, $\delta^{18}O$ and dust concentration, and three other derived pollen records (see below), to produce an annual-layer-counted timescale from the 2011 glacier surface to 53 m (41 m w.e.) depth. Note that StratiCounter also produces an age uncertainty interval for the resulting timescale. Below 53 m (41 m w.e.) depth, where the StratiCounter analysis became more uncertain, we attempted visual counting down to 57 m (45 m w.e.), where the combined annual signals seemed distinguishable (Fig. S4-6).



The additional three pollen records used were: i) Day Of the Year, which considers 32 pollen types (DOY32) or; ii) 46 types (DOY46) with known blooming time and; iii) a depth-to-day-match of the pollen compositional record obtained using 10 years of data collected at the nearby Bolzano airborne pollen monitoring station (~70 km from the drilling site; https://ean.polleninfo.eu/Ean/). The three pollen records were weighted as a single line of information by StratiCounter as they

5   are not independent. Briefly, each of these three records was derived from the different classified pollen types and provided an estimate of the pollen ensemble depositional day-of-year (DOY; values between 1 and 365) based on the observed pollen type spectra composition in the ice core (for details please see Festi et al., 2017).

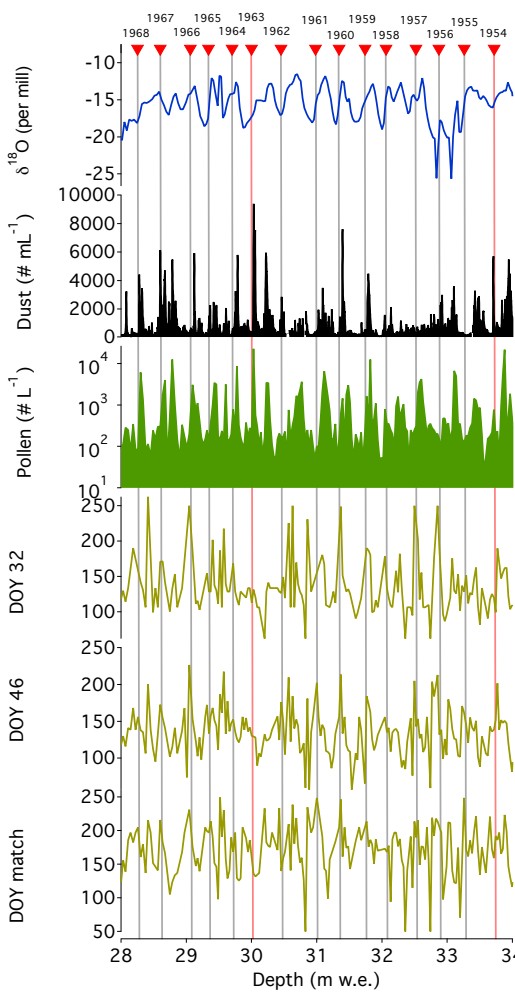

10   Figure 4: Annual layers between 28 and 34 m water equivalent (w.e.) in core #1 shown in $\delta^{18}$O, dust and pollen concentrations, DOY 32 and 46 and DOY depth-to-day match variations (see text). Red vertical lines indicate time markers from the $^3$H and beta activity peaks (Gabrielli et al., 2016). See also supplementary figs. S4 and S5 which display annual layers in previous periods during the 20th century.



| Core # | Depth #1 (m w.e.) | StratiCounter original timescale (CE) | Adjusted age (CE) | Time marker age (CE) | Time marker |
|---|---|---|---|---|---|
| 1 | 5.98 | 2005.9 | 2006.1 | Spring 2006 | Spruce extreme |
| 1 | 13.84 | 1994.9 | 1995.1 | Spring 1995 | Spruce extreme |
| 1 | 18.27 | 1986.6 | 1986.8 | 1986 | Beta activity (Chernobyl) |
| 2 | 29.9 | 1960.3 | 1962.5 | 1963 | Beta activity, $^3$H |
| 2 | 33.89* | 1952.7 | 1954.9 | 1955 | Beta activity |

*Tranferred from core #2

Table 3: Time markers in the upper portion of the Alto dell'Ortles cores, which were also used to adjust the StratiCounter annual layer counting by two years.

The derived annual layers were compared and subsequently adjusted within the derived timescale uncertainty to independently fixed time markers that were identified in core #1 (Table 3). These include three known horizons with high levels of radioactivity: the beta activity peak from the Chernobyl accident in 1986; beta and $^3$H peaks from atmospheric nuclear tests attributed to 1963; and a first beta intensity peak from detectable radioactive fallout in 1955 (Gabrielli et al., 2016; Gabrieli et al., 2011). Two additional time markers were assigned by comparing spruce (*Picea abies*) pollen concentrations with

observations from the nearby pollen measuring stations at Innsbruck and Obergurgl (Bortenschlager and Bortenschlager, 2003). The records from these stations are the longest (44 and 41 years, respectively) in this area and contain evidence of three exceptional blooming years in 2006, 1995 and 1992. While the strongest episode in 1992 does not correspond to large spruce concentrations in the Alto dell'Ortles core, the 1995 and 2006 events are characterized by high and broad pollen peaks in the firn portion of the core. The ages of all the marker horizons fall within the derived uncertainty of the StratiCounter timescale.

By using these five time markers as guides for adjustments, we removed 2 years between 18.2 and 33.89 m w.e. thereby producing the final annual-layer-counted timescale.

     We compared the annual timescale obtained by StratiCounter with the $^{210}$Pb ages derived from the upper part of core #2 that were used to constrain TC2016, which was not based on annual-layer counting (Fig. 5b and S6). The two independent time scales are in good agreement down to 53 m (41 m w.e.; ~1940 CE), while they diverge below. This offset can partly be

explained by the fact that the initial dating by $^{210}$Pb did not account for any layer thinning, statistically not evident from the activity concentrations. Additionally, a slight offset in the $^{210}$Pb ages due to imprecise determination of the background of $^{210}$Pb cannot be excluded. A plateau in activity concentration below a certain depth, clearly indicating the $^{210}$Pb background level (Gäggeler et al., 2020), was not observed and its determination was based on one single data point only. In contrast, the extended visual annual layer counting from 53 m (41 m w.e.) to 57 m (45 m w.e.) connects well with the most recent

synchronization ties of the Alto dell'Ortles and AN Pb records (see below and Fig. 7c, S6). In conclusion, because of the much higher resolution, annual layers were adopted to replace the $^{210}$Pb time markers initially used in TC2016.



## 4 COPRA fitting of the dating horizons

In line with the development of TC2016, we built a revised continuous depth-age relationship of the Alto dell'Ortles cores (Fig. 5) by fitting all the empirical time markers (defined in m w.e. depth and years before 2012; Supplementary Table 1) within their linked uncertainty ranges by means of Markov chain Monte Carlo simulation (COPRA model, 2000 simulation runs). This provides a depth-age relationship with a linked depth-time dependent uncertainty (Breitenbach et al., 2012). For the most recent portion of the chronology (2011-1927), the original uncertainty of the single annual layers provided by StratiCounter (1–10 years) was retained. Overall, 113 annual time markers provided by StratiCounter from 2012 to 1900, anchored by five distinct time horizons (Table 3) were combined with seven [14]C dated layers in the deep part of the record (651 CE back to around 7000 years before present) (Fig. 5).

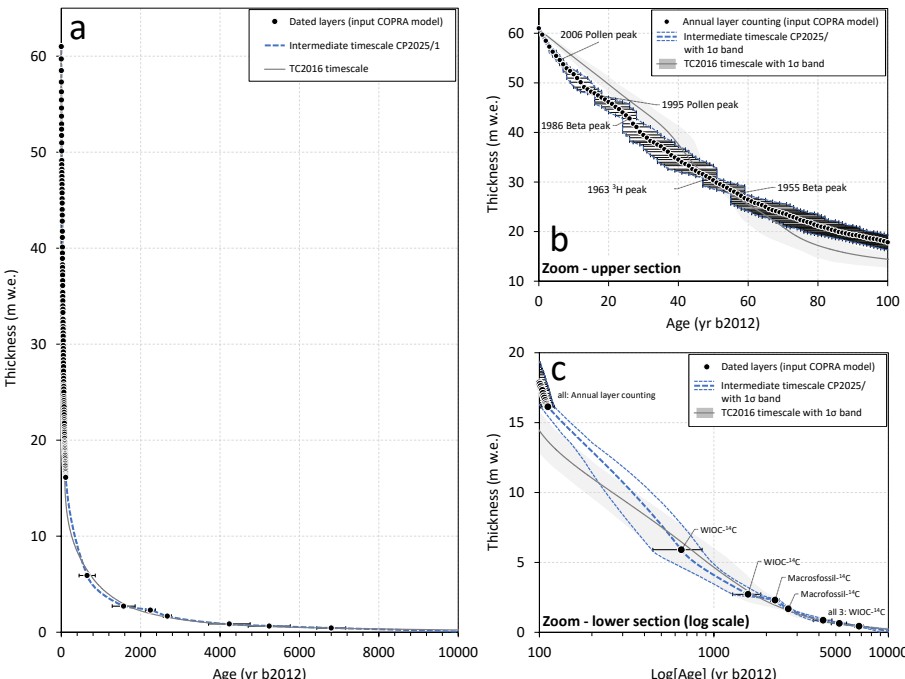

Figure 5: Comparison of the TC2016 and the updated, intermediate Alto dell'Ortles (CP2025/1) timescales obtained by using COPRA. The different time markers for construction (see Supplementary Table 1) are indicated by symbols with error bars representing their uncertainty by the linked dating methods (1σ range for [14]C cal ages; see text for others). Shown in panel (a) for the entire length of the ice core from surface, indicated by the uppermost point, to bedrock covering the entire period of the Alto dell'Ortles ice archive, in (b) for the last 100 years, and in (c) for the older part of the record (>100 years) on a logarithmic age scale.



Comparison of this updated Alto dell'Ortles timescale (CP2025/1) with the previously published TC2016 chronology is displayed in Fig. 5. The two timescales differ significantly during the most recent 20 years, as a result of replacing the $^{210}$Pb time markers by the annual layers (also taking into account three additional time markers of 2006, 1995, 1986; Table 3; Figures 5b and S6). Other than the most recent time interval, CP2025/1 and the TC2016 chronologies essentially agree within their uncertainties (Fig. 5). However, CP2025/1 is 20-100 years older between 1920 and 1600 CE.

## 5 CP2025/1 synchronization with a well-dated millenial Arctic Pb record (171 BCE - 1900 CE)

Using Pb records for synchronizing ice core chronologies is a recently developed approach (Preunkert et al., 2019; Osman et al., 2021) that we adopt in this study to further refine CP2025/1. In general, well-dated Pb records from six Arctic cores (McConnell et al., 2019) reaching back to 200 BCE show similarities with the high-resolution Pb record from the Alto dell'Ortles ice core (Fig. 6). Among the Arctic Pb records, the one from the AN ice cap in Severnaya Zemlya, Russian Arctic was selected for synchronization with the Alto dell'Ortles cores (Fig. 6) because: i) it contains one of the longest continuous chronologies; ii) it provides the largest Pb amplitude signal; and, iii) it is suggested by atmospheric modelling to be the most influenced by Pb emissions from silver mining and metallurgical activities in central Europe close to Mt. Ortles during the past millennia (McConnell et al., 2019), making it more likely that the Alto dell'Ortles and AN drilling sites share similar variability in atmospheric Pb deposition. The uncertainty of the AN chronology was estimated to be less than 5 years between 500 and 1999 CE, with less than 2 years uncertainty at the known volcanic time horizons within the Greenland reference record (McConnell et al., 2019).

The continuous flow analysis Pb record from Alto dell'Ortles core #3 was selected for synchronization with AN because of its higher time resolution. In this case the non-crustal Pb concentration was obtained by using the Pb/Rb ratio of 0.51 (value obtained from a deep core #3 section characterized by the lowest Pb concentrations between 69.55 and 69.94 m depth). To match the non-crustal Pb concentration between AN and Alto dell'Ortles on the intermediate CP2025/1 timescale, synchronization was performed using the Analyseries 2.0.8 software (Paillard et al., 1996) (see below), with the records being averaged to a 10-year time resolution which was used for matching the midpoints to avoid over-interpretation of individual data points and to account for the dating uncertainty of the AN dating reported (up to ±5 years). This approach also allowed a better estimate of reasonable tie-point age uncertainties.

Core #3 was synchronized to AN by using the 11 most prominent features of the two non-crustal Pb concentration records between 175 BCE and 1755 CE (Fig. 6). No synchronization tie-points were selected for the more recent period for which the Ortles ice cores were dated based on the annual layer counting in core #1 (the two records on their respective, independent timescales already showed high correlation; Pearson's r of 0.61 for annual values, p<0.001). Note that, as a consequence, the final Alto dell'Ortles records remained on their independent timescales for the 20th century. For the selected tie-points, the shift in age (CP2025/1 - AN age) was ~10-20 years or less back to 1600 CE; around 40 years back to 1550 CE, between 120 and 270 years from 1250 to 250 CE; and between -280 to -240 years between 0 and 200 BCE. Importantly, 7 out



of the 11 tie points were within the 1-sigma uncertainty of the initial CP2025/1 timescale built based on the absolute ages from [14]C, with the other 4 being within the 2-sigma range (see figs. 6 and 7). This ensured a quantitatively controlled and robust wiggle matching.

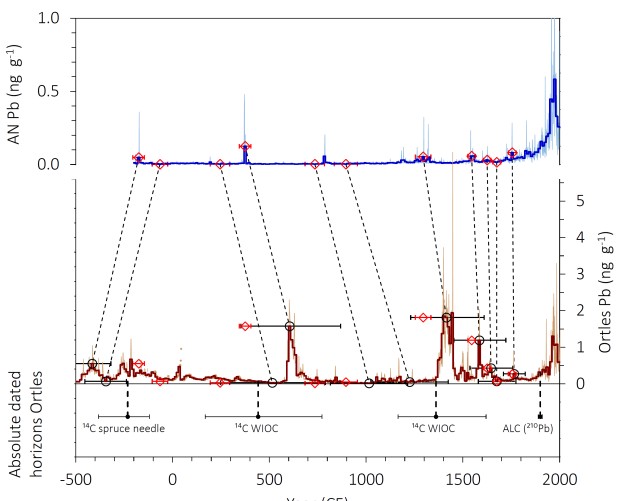
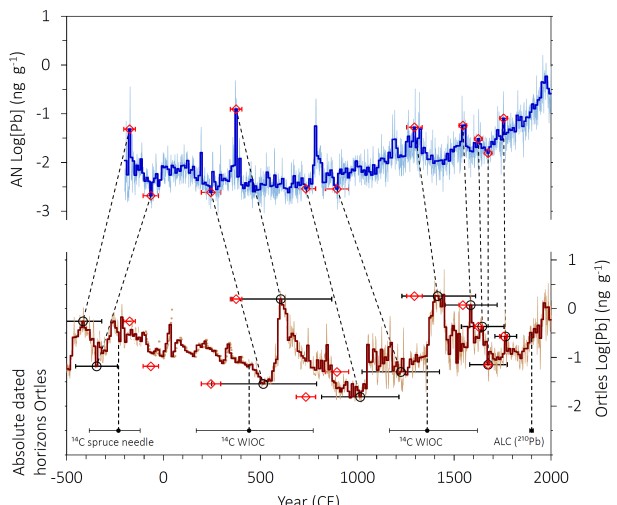

Figure 6: Synchronization of the Pb record of non-crustal concentrations from the Alto dell'Ortles core #3 (dark red record) on the CP2025/1 timescale with the record from Akademii Nauk (AN; blue record) on its well-dated (±5 years) chronology used as reference (McConnell et al., 2019). Thin and thick lines show annual and 10-year averages, respectively. The tie points selected for synchronization by wiggle-matching are connected by the thin dashed lines and indicated by red diamonds (AN; error bars indicating the age uncertainty of the selected tie points) and black circles (Alto dell'Ortles; error bars indicating the 1σ uncertainty linked to CP2025/1). Absolute ages from [14]C dating (or annual layer counting) are shown at the bottom (black dots indicate the μ-age and error bars in the 1σ range), with the thick black dashed line connecting to the record for visualizing the time horizons used to construct CP2025/1. Same for both panels, except for Pb concentrations being plotted on a logarithmic scale on the right panel.

## 6 Final Alto dell'Ortles Age Scale CP2025/2

The 113 annual time markers provided by counting annual layers from 2012 to 1900 (sect. 3.2), the 11 tie points obtained from the match of the Alto dell'Ortles core #3 with the AN Arctic Pb record (1755 CE-175 BCE; sect. 5) and the five [14]C ages from the deepest part below (659 BCE back to 7000 years; sect. 3.1) were finally used to develop the final Alto dell'Ortles time scale CP2025/2. Using these data points (with depths in m w.e. and the age as years before 2012; see summary in Supplementary Table 2), the exact same methodology as applied to construct TC2016 and CP2025/1 was applied (Markov chain Monte Carlo simulation, 2000 simulation runs, using COPRA, Breitenbach et al., 2012) fitting the time markers with



their given 1σ uncertainty. The resulting final Alto dell'Ortles depth-age relationship, COPRA25/2 with its linked uncertainty is shown in Fig. 7.

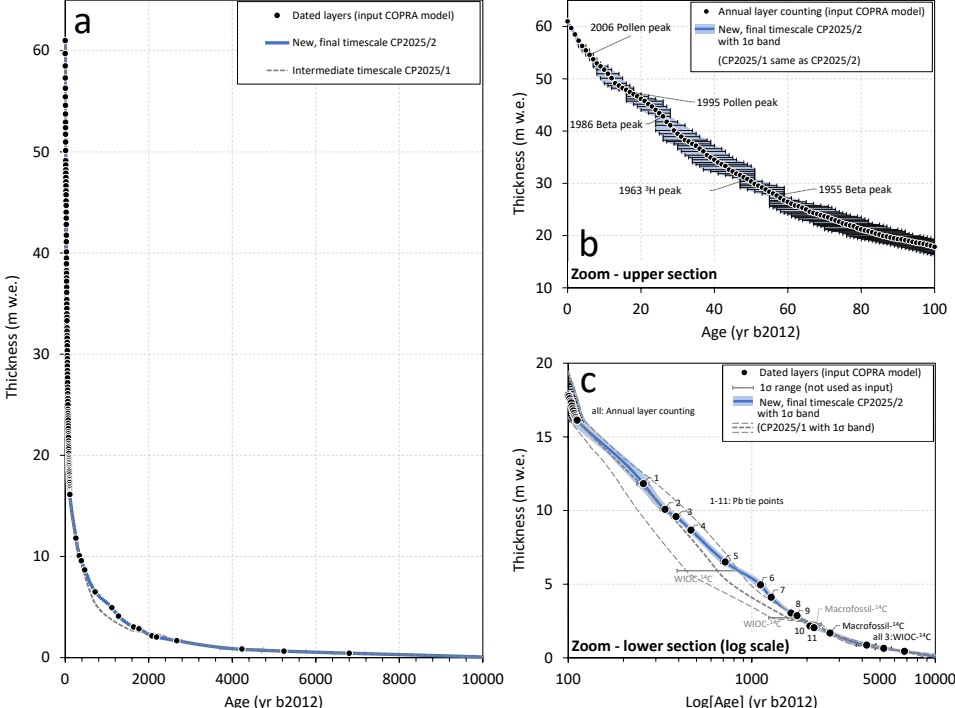

Figure 7: Comparison of the intermediate (CP2025/1, before synchronisation) and the final Alto dell'Ortles (CP2025/2) timescales. The different time markers used for construction (see Supplementary Table 2) are indicated by symbols with error bars illustrating their uncertainty by the linked dating methods (1σ range for $^{14}$C cal ages; see text for others). The two $^{14}$C data points that were not used for model input are indicated just by their 1σ range (grey bars). Shown in panel (a) for the entire length of the ice core from surface, indicated by the uppermost point, to bedrock covering the entire time period of the Alto dell'Ortles ice archive, in (b) for the last 100 years, and in (c) for the older part of the record (>100 years) on a logarithmic age scale.

With the final time scale at hand (CP2025/2), we can assess the synchronisation, comparing the Alto dell'Ortles non-crustal Pb concentration records with the corresponding records from AN (Fig. 8) and the independently dated, slightly shorter Pb record from Colle Gnifetti in the Western European Alps (CG03B, Switzerland; Supplementary Text 2 and figs. S1-S3). In general, the match between Alto dell'Ortles and AN is remarkable and provides additional evidence of the stratigraphic and



temporal continuity of this ice core record (i.e., no evidence of age reversals or ice flow disturbance). By applying the synchronization, the correlation over the entire record increased from 0.26 to a significant value of 0.44 (r, p < 0.001), for the record before 1750 CE from 0.09 to 0.46 (r, p<0.001). Also, the agreement between Alto dell'Ortles on CP2025/2 and the fully independently dated CG03B ice core is remarkable back to ~1500 CE (r=0.41, p<0.001) (Fig. S2). We also note that while

5 CG03B agrees well with AN back to ~1400 CE, the records/chronologies start to diverge further back in time (the uncertainty of the CG03B dating increases from 7 years at 1763 CE to around 300 years for the year 0 CE/BCE; Fig. S1). This comparison between CG03B and AN further corroborates the idea of a common Pb atmospheric signal in Central Europe and the Arctic in the past when emission sources were sparse (also see McConnell et al., 2025).

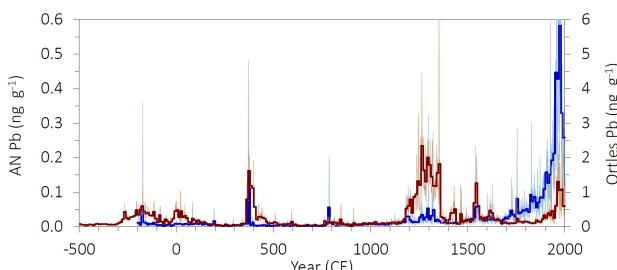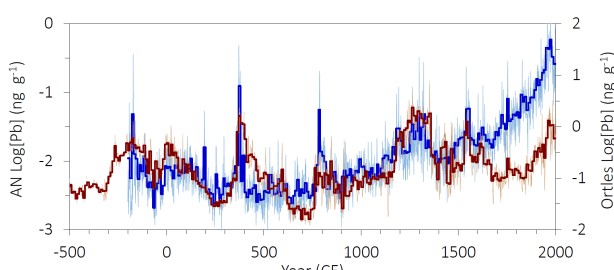

Figure 8: Comparison of the non-crustal Pb concentrations from Alto dell'Ortles #3 (Ortles; dark red) after synchronization with Akademii Nauk (AN) used for age reference between 175 BCE and 1750 CE. Thin and thick lines show annual and 10-year averages, respectively. Same for both panels, except for Pb concentrations being plotted on a logarithmic scale on the right panel.

## 7 Investigation of the empirical timescale CP2025/2 by glaciological ice flow modelling

We applied the Dansgaard-Johnsen (DJ) glaciological ice flow model (Dansgaard and Johnsen, 1969) to investigate the revised age-depth relationship CP2025/2 at the Alto dell'Ortles drilling site (see also Supplementary Text 3). The DJ model (one-dimension; 1D), uses an approximation of the ice flow to calculate the vertical strain rate (thinning of annual ice layers with depth). It results from assuming the horizontal velocity ($v_x$) described by Glen's law to be constant from the glacier surface down to a given height above bedrock ($h$; shear zone thickness), where it decreases linearly with depth to zero at the bed (no-basal ice flow, i.e., ice frozen to bedrock; blue curve in Fig. 9). It should be noted that the term h is the result of the DJ model-approximation for $v_x$ and subsequent mathematical reformulations, rather than being descriptive of a distinct separation between two zones existing in reality.

25 A general limitation of modelling the age-depth relationship, independent on the complexity of the applied ice flow model, arises for the assumption of steady-state conditions. Unless additional glacier mass balance related information far into the past were available, this assumption is unavoidable. However, information about the age of a layer at a certain depth can provide invaluable constraints for model parameters that are not accessible otherwise (equally for models of lower or higher



complexity). For reasonable and meaningful model output, the choice of parameter values is crucial. Here, we used two independent approaches (A1 and A2) for the determination of annual net accumulation rate ($b$) and $h$. A1 is based on present-day glaciological observations (section 7.1), and (A2) is based on the time information provided by the empirical dating described in this study (section 7.2). We further used parameters from A1 in a mixed approach (B) with A2 (section 7.3 and

Table 4). In all cases steady-state conditions were assumed, which for the applied model (DJ) can be defined as: i) constant glacier thickness ($H$), ii) no-basal ice flow, iii) constant annual net accumulation rates (b), and, consequently, iv) a constant shear zone thickness (h). For the glacier thickness $H$ and the condition of non-basal flow, we relied on present-day observations. For $H$, we adopted the present value of 74.88 m (61.15 m w.e.) (Gabrielli et al., 2016), implying that the ice thickness had not changed significantly since glacier build-up at the drilling site ~7 kyrs BP. The "no-basal ice flow" assumption seems

reasonable considering the ice temperature at bedrock of -2.8 °C in 2011 (Gabrielli et al., 2012), despite the recent warming.

## 7.1 Approach A1: determination of *b* and *h* based on present-day glaciological observations

For $b$, the value of 1.0 m w.e. y$^{-1}$ as derived from field observations was considered as a best available estimate (Gabrielli et al., 2016; Festi et al., 2015). For the determination of $h$, we took advantage of the available data from englacial

cumulative displacement measurements obtained with an inclinometer in fall 2011 (borehole #2; see Fig. 4 in Gabrielli et al., 2016). The vector sum of the annual displacement along the two spatial axis (X, Y) yields a direct measure of $v_x$ for each depth (relative to the lowermost layer measured). Because the DJ model requires units to be in m w.e. (see Supplementary Text 3), we transformed the original displacement data (in m) accordingly, using the measured smoothed density profile derived from the ice core (Fig. 9). The smoothing thereby accounts for spatial inhomogeneities in firnification (with relevant space, due to

ice flow, assumed to exceed the ice core cross section). The profile indicates a clear imprint/variation related to the transition between the characteristic Alto dell'Ortles zones (Fig. 9): (i) the active layer, with a displacement signal observed elevation above the bedrock of ~52 m w.e., which is in agreement with determination based on englacial temperature measurements (personal communication Roberto Seppi), and (ii) the temperate zone, within 0.5 m of the depth previously reported (Gabrielli et al., 2016). Notably, the profile of horizontal displacement per year (i.e., $v_x$) allows determination of a best estimate value for

the shear zone height $h$ of 37.3 m w.e. (obtained by the method of least residual sum of squares between measured and approximated $v_x$). The degree of consistency between the DJ model approximation of the $v_x$ vertical profile and the actual measurements is remarkable (see Fig. 9). We consider the observed consistency as a strong argument for the applicability of the DJ ice flow model for the Alto dell'Ortles drilling site and likely also for other alpine glacier settings characterized by strong layer thinning near bedrock.




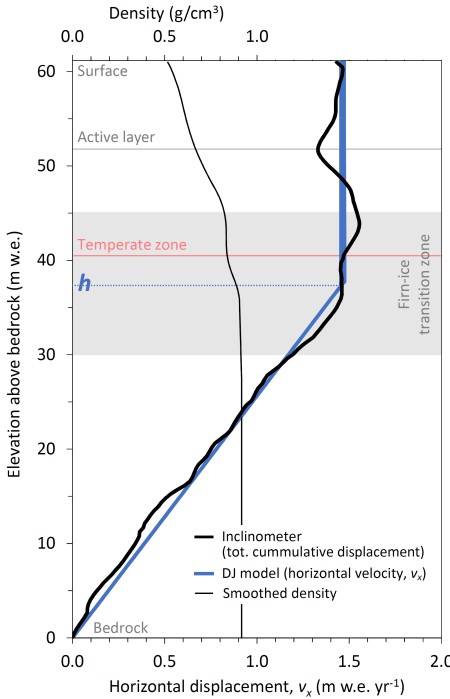

Figure 9: Vertical profiles of density and horizontal displacement (i.e., $v_x$) for the Alto dell'Ortles drilling site. Also indicated is the elevation above bedrock defined as shear zone thickness in the DJ model (parameter $h$), which was derived from the displacement measurements (see main text). The firn-ice transition zone, typically with density of 0.830 - 0.917 g cm⁻³, (Cuffey and Paterson, 2006) is indicated by the grey shading while the grey and red lines mark the bottom of the active layer and temperate zone, respectively (see main text).

**7.2 Approach A2: determination of *b* and *h* based on time information provided by the empirical dating**

In this case, both $b$ and $h$ were treated as free model parameters. The best set of values was then derived from minimization of the misfit between the DJ modelled ages and those of the empirically dated layers provided in Supplementary Table 2 (least sum of time-weighted squares of residuals; Fahnenstock et al., 2001). This approach results in values of 1.09 m w.e. y⁻¹ and 33.8 m w.e. for $b$ and $h$, respectively.

**7.3 Approach B: combination of approaches A1 and A2**

Either $b$ or $h$ was pre-set as derived in A1 while the other parameter was kept free, with its value then derived following the minimization methodology described for A2. By prescribing $b = 1.0$ m w.e. yr⁻¹, the value of $h$ with the least misfit is 27.7 m w.e. (approach Bb). By prescribing $h = 37.3$ m w.e., the resulting value for $b$ is 1.13 m w.e. yr⁻¹ (approach Bh).



### 7.4 Ice flow model results, interpretations and limitations

The model parameter values, either inferred from present-day observations (A1) or based on information related to time (ice core) (A2), lay within a relatively narrow range (1.0 - 1.13 m w.e. yr$^{-1}$ and 27.7 - 37.3 m w.e. for $b$ and $h$, respectively). This is remarkable considering that:

i.    for A1, $b$ is based on 8 years of observations while $h$ presumably reflects flow conditions of a significantly longer time span due to delayed glacier response to changing conditions, see e.g. Cuffey and Paterson, 2006.

    ii.    for A2, $b$ and $h$ reflect their corresponding averages for a time-period of ~7000 years.

    iii.    While $b$ and $h$, as determined by A1, are due to glacier response times likely not directly related, A2 highlights how empirically derived time markers allow reasonable and related estimates of model parameters (even if recent

glaciological observations are lacking). By constraining parameters $b$ and $h$ with time information (approaches A2, B) model results, fulfilling the equation of continuity and respecting mass conservation, an agreement with empirical dating can be achieved (see section 7.1). This is illustrated by the fact that the time span between the lowermost dated layer by $^{14}$C and the surface (6383-7227 years b2012, 1σ uncertainty range) is largely consistent with the model result using the sets of parameters from A2 (6192 years b2012)

iv.    years) and B (5933 and 6340 years b2012 for Bb and Bh, respectively). Here, B further provides insights into robustness and sensitivity of A2, demonstrating how a difference in the estimate of a single model input value requires adjustment of at least one of the other model parameters, if the time span contained in the ice column is fixed.

    v.    The modelled timescales obtained by using the different sets of parameters are displayed in Fig. 10 (A1, A2 and Bb; Bh not shown as it is visually indistinguishable from A2 and Bb). The relatively close range of values derived for the

parameters ($b$, $h$) and the agreement of the modelled timescales obtained from A1, A2, and B, suggest that over the entire period covered by the ice core, $b$ (and accordingly $h$), on average, must have been close to the modern observed values. While for A1 the modelled time span contained in the ice column (7184 years) is in good agreement with the absolute dating by $^{14}$C, this is not the case for the last ~100 years, where the model results for the scenarios A2, Bb and Bh show better agreement (see Table 4).

| Scenario | $h$ (m w.e. above bed / depth) | $h$ determination | $b$ (m w.e.) | $b$ determination | Age at 45.02* m w.e. (yrs b2012) | Age at 60.71** m w.e. (yrs b2012) |
|---|---|---|---|---|---|---|
| A1 | 37.3 / 23.8 | present day measurement | 1.00 | present day measurement | 147 | 7184 |
| A2 | 33.8 / 27.3 | tuned to fit dated layers | 1.09 | tuned to fit dated layers | 127 | 6192 |
| Bb | 27.7 / 33.5 | tuned to fit dated layers | 1.00 | present day measurement | 126 | 5933 |
| Bh | 37.3 / 23.8 | present day measurement | 1.13 | tuned to fit dated layers | 129 | 6340 |
| C | 35.0 / 26.2 | picked for illustration | 0.60 | picked for illustration | 237 | 11534 |

*Depth to where annual layer counting could be performed, resulting with an age of 112±10 years before 2012.

**Mid-depth of the lowermost dated layer with a calibrated 1 sigma $^{14}$C age range of 6383-7227 years before 2012.

*** Present day determination for total thickness H is kept constant ( 61.15 m w.e.) in all the scenarios

Table 4: Summary of parameters and results for the different ice flow modelling scenarios. See main text, and figs. 9 and 10 for additional details.

Reasonable good agreement is obtained for all of them between around 7000-4000 years b2012. However, for the intermediate part (4000-100 years b2102) all approaches fail to reproduce the observations, underestimating the age of the ice by several decades to about one millennium in the upper and lower portion of this section, respectively (Fig.

5      10). This significant discrepancy suggests that conditions were different notably from the middle Holocene until the end of the Little Ice Age (LIA; 1250-1850 CE) (Pages K Consortium, 2013). The most likely explanation is a lower annual net accumulation occurred during that time (also see below).

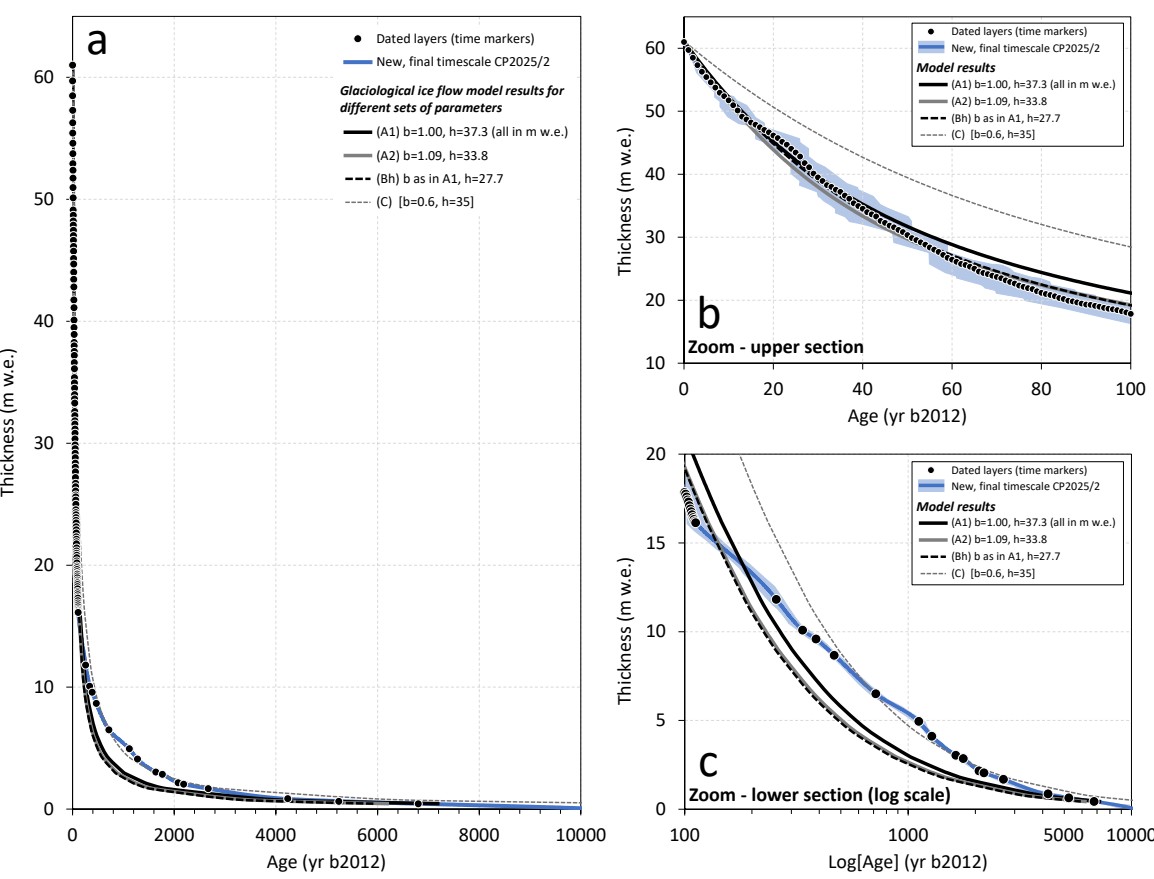

Figure 10: Comparison of the revised, final empirical timescale for the Alto dell'Ortles ice cores (CP2025/2) with several runs

10     of the Dansgaard-Johnsen glaciological ice flow model for the indicated sets of parameter values (see text for details). Shown in panel (a) for the entire length of the ice core from surface, indicated by the uppermost point, to bedrock covering the entire period covered by the Alto dell'Ortles ice archive, in (b) for the last 100 years, and in (c) for the older part of the record (>100 years) on a logarithmic age scale.



To investigate the hypothesis of changes in net annual accumulations rates over that period, we used pre-set, hand-picked values for *b* and *h* for the model input (approach C, Fig. 10, Table 4). For *b* (0.6 m w.e. yr$^{-1}$) and *h* (35 m w.e.) at least a partial match of the DJ model with our revised empirical timescale was obtained for the intermediate section of the core (Fig. 10). Thus, while the steady-state assumption for modelling can be valuable in order to estimate the time span of the ice archive (e.g. Lüthi and Funk, 2000) it seems invalid to derive an accurate age-depth relationship based on our results, at least for the Alto dell'Ortles drill site. Here we note that the selected value of *b* = 0.6 m w.e. yr$^{-1}$ is not constrained in any way, and can thus be considered qualitative only, to illustrate a possible change in net accumulation for the discussed period. To derive a quantitative estimate of *b*, a non-steady-state approach under consideration of continuity would be required (inverse modelling, see below).

Our interpretation of the ice flow modelling suggests lower snow accumulation during the central period of the Holocene (100-4000 years b2012), particularly during the LIA. A change in conditions for the depth interval corresponding the LIA, recorded right below ~57 m depth (or ~16 m w.e. above bedrock; Fig. 9), is consistent with intensification in visible ice layer thinning in the Alto dell'Ortles ice core cores (see Fig. 7b in Gabrielli et al. 2016) which is concomitant with a sudden increase in the frequency of the δ$^{18}$O signal (Fig. 1). As different paleoclimate archives indicate higher precipitation in the Alps during LIA (Magny et al. 2011 and references therein), a lower snow accumulation on Alto dell'Ortles may be explained by a more efficient wind erosion of colder and light (low density) surface snow due to lower air temperatures, particularly during the warm seasons from late spring through fall. If modern temperatures are the warmest since the Northern Hemisphere Climate Optimum, it is possible that this hypothesis could hold from the beginning of the middle Holocene (4000 years ago) until the end of LIA (19$^{th}$ century).

Our suggestion of variations in snow accumulation over time is not unprecedented for high altitude glaciers. In the few low-latitude drilling sites where it was possible to measure changes in past snow accumulation by layer counting (Thompson et al., 2000; Winski et al., 2017) or based on $^{14}$C ages (Herren et al., 2013), these observations implied non-steady-state conditions which impacted the timescale. While inverse modelling approaches that consider non-steady-state conditions and take advantage of empirically derived time markers are well established for polar drilling sites (e.g. Buiron et al., 2011; Buchardt and Dahl-Jensen, 2008), efforts in this direction are also clearly needed for glaciers from high altitude/low latitude sites.

## 8 Conclusions

An intermediate revised chronology (CP2025/1) was obtained for the Alto dell'Ortles ice cores by realigning the depths of core #1, #2, #3 and incorporating an absolute time marker from a newly discovered $^{14}$C dated macro-organic fragment (232 ± 126 BCE); and new data points from counting annual layers (from pollen, dust and δ$^{18}$O records) visually and by means of an automatic algorithm (StratiCounter) between 1900 and 2011. All the time markers were combined and fitted for a continuous depth-age scale by Markov chain Monte Carlo simulation to obtain an improved chronology compared to the



previously published version (TC2016), which only relayed on absolute dates from [210]Pb in the youngest part (1900-2012 CE), resulting in more accurate chronology particularly for this section. By synchronization of the Pb concentration records from the Alto dell'Ortles Pb ice core records (adopting CP2025/1) with a well-dated polar ice core record from Akademii Nauk (AN, Russian Arctic) used as the age reference from 1750 CE back to 175 BCE, 11 additional tie points for model input were

selected and combined with the already available time markers (outside the synchronization time range). Therefore, with this iterative approach, the subsequent age model output yielded the final, fine-tuned new Alto dell'Ortles chronology CP2025/2.

The new [14]C age (232 BCE; stratigraphically consistent with the previous [14]C dates), and the correlation between the non-crustal Pb CP2025/2 and AN records further indicates that stratigraphic disruption caused by disturbance in ice flow can be ruled out, at least for age inversions exceeding the current dating uncertainty and the resolvable spatial sampling resolution.

Finally, an ice flow model investigation of CP2025/2 was performed by adopting a simple 1D Dansgaard-Johnsen model, the results of which suggest that non-steady state conditions (e.g. changes in snow accumulation rate over time) need to be considered to explain the resulting CP2025/2 age-depth relationship. This particularly concerns the Little Ice Age (1250-1850 CE) when a decrease in snow accumulation rate was inferred, perhaps because of an increase of wind erosion efficiency. Clearly, non-steady state approaches that are currently used in polar regions should be considered and applied to model the ice

flow at high altitude-low latitude ice core drilling sites.

In conclusion, this revised more accurate time scale (CP2025/2) will allow to provide new detailed climatic environmental histories of Central Europe during the Holocene. In addition, CP2025/2 has the potential to become a reference chronology for multiple paleoclimate archives in Europe.

**Data availability:** The data presented in this work are archived at the National Oceanic and Atmospheric Administration World Data Center-A for Paleoclimatology: (*link when available*).

**Author contribution:** PG and TJ designed the structure of the revised chronology and wrote the paper; MB, PG and CB performed the continuous flow analyses of Pb; DF, WK and KO performed the pollen analyses and their interpretation; MW

and DF adopted the StratiCounter algorithm to count annual layers; GD and BS studied the stable isotopes data set; PG, TJ and MB performed the matching of the ice core records; TJ designed/run the ice flow model experiments with initial input from MW and evaluated the results with PG and MS. All the authors discussed and reviewed the manuscript.

**Competing interests:** Barbara Stenni and Carlo Barbante are members of the editorial board of Climate of the Past.


**Acknowledgments**

This work is a contribution to the Ortles project, a program supported by NSF awards #1060115 and #1461422 to The Ohio State University (OSU) and by the Fire Protection and Civil Division; Südtirol, Abteilung Bildungsförderung, Universität und Forschung of the Autonomous Province of Bolzano - Südtirol in collaboration with the Forest Division of the



Autonomous Province of Bolzano - Südtirol and the National Park of Stelvio. We are grateful to: Giuliano Bertagna and Ping Nan Lin for performing Pb and $\delta^{18}$O analyses at OSU; and John Bolzan, Ian Howat, Luca Carturan and Roberto Seppi for useful discussions linked to the modelling section. We are also thankful to Joe McConnell and an anonymous reviewer for providing constructive suggestions that allowed to improve an earlier version of our manuscript. This is Ortles project

publication 10 (www.ortles.org).

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
