# Peer review of "A multimillennial Alpine ice core chronology synchronized with an accurately dated Arctic Pb record"

_EGUsphere, 2025_

## Referee Comment (RC2)

**Review of A multimillennial Alpine ice core chronology synchronized with an accurately dated Arctic Pb record, by Paolo Gabrielli et al.**

A preliminary absolute timescale of a low latitude-high altitude Alpine ice core drilled in 2011 at the glacier Alto dell'Ortles (3859 m, Eastern Alps, Italy) was obtained in 2016 based on a peak in $^3$H activity, $^{210}$Pb, and $^{14}$C dating, indicating that the record spans the last ~7000 years (Gabrielli et al., 2016). The present work improves this preliminary dating based on additional information. First, $^{14}$C dating of a fragment of a charred spruce needle present in the basal ice provided an age (232 ± 126 BCE) which agrees with previous $^{14}$C dates in the oldest part of the record. Second, novel seasonally resolved pollen records from the upper firn/ice portion of the Alto dell'Ortles cores were combined with $\delta^{18}$O and dust annual variations to refine the dating for the 20$^{th}$ century. Finally, the Pb Ortles records were used to match the depth scale of two of the Ortles cores and with a Pb record from an Arctic ice core (AN), well-dated (±5 years) for the ~200 BCE to ~2000 CE period.

**Summary (for details see overall and specific comments below) of aspects accounted for**
- Does the paper address relevant scientific questions within the scope of CP?
    If geochemically discussion of Pb records will be addressed in a revised version (see comments below), yes, if not, no.
- Does the paper present novel concepts, ideas, tools, or data?
    No, except for Pollen analysis
- Are substantial conclusions reached?
    Not really, but already existing ice core dating was improved.
- Are the scientific methods and assumptions valid and clearly outlined?
    No, see comments below
- Are the results sufficient to support the interpretations and conclusions?
    No, see comment below
- Is the description of experiments and calculations sufficiently complete and precise to allow their reproduction by fellow scientists (traceability of results)?
    No, see comments below
- Do the authors give proper credit to related work and clearly indicate their own new/original contribution?
    Yes, but some references are missed
- Does the title clearly reflect the contents of the paper?
    No since comparison with Arctic is not geochemically discussed
- Is the overall presentation well structured and clear?
    Yes, but analytical section is missing
- Is the language fluent and precise?
    Yes
- Should any parts of the paper (text, formulae, figures, tables) be clarified, reduced, combined, or eliminated?
    Yes, see comments below

**Overall comment**:
Whereas the improvements of the preliminary dating of the Ortles records based on the new $^{14}$C dating, the pollen records, and the use of a 1-D ice flow model used to test steady state conditions of the glacier, are adequately presented and surely merit to be published, the

presentation of the Pb records from the Ortles ice cores is made rather poorly: There is a) an absence of analytical details, b) a not very serious discussion on the difference between the Pb records of Ortles core 1 and core 3, and c) an absence of a geochemical reasoning for the validity of the comparison between the Ortles and the AN Pb record. On this last point, on the one hand, the authors argue that the geochemical discussion of the Ortles Pb records are out of the scope of this paper, but on the other hand they adjust two Ortles Pb peaks (dated by Ortles [14]C to environ 400 and 230 BC) on the two AN peaks attributed by Mc Connell et al. (2019) to the roman Republic and Empire (70 BC and 100 CE), without clearly specifying in the text that this fit is based on the assumption that Ortles ice has recorded the Roman antiquity, and without discussion on the geochemical reason for this assumption, respectively. This is incomprehensible and incoherent already (see further comments below) because there is a potential alternative interpretation of this two peaks in the Ortles Pb record which leaves them at the age attributed by the [14]C and which is not presented in the manuscript. Also, there are major differences between the Ortles and AN Pb records (see Figure 6 and 8), e.g. from 400 to 700 CE and after 1400 CE to recent times, which strongly require geochemical comments. Given also numerous errors, the very poor quality of Figures reporting on Pb (some are completely unreadable), the non-convincing explanation for differences of the Pb records in core 1 and 3, and finally the quasi-absence of a plus value of the comparison between Ortles and AN (if correct), I strongly recommend to refocus the paper on the other dating improvements only and skip the whole presentation of Pb. If the authors decide to maintain the Pb part of the manuscript, they have to provide a complete geochemical discussion (and revise the text with respect to several problems that will be detailed below).

**Specific comments:**

1) Abstract:
Whereas at the time of submission, the authors could not have known about recent findings made in the French Alps at the ice core drill site of Dome du Goutier (DDG) showing undisturbed continuous climate and aerosol ice records spanning the last 12,000 years (Legrand et al., 2025), however this new finding should be referenced and the wording in the text adequately updated.
For instance, the first sentence of the abstract (line 20) is not true anymore: please change to "provided evidence that the record spans the last 7,000 years" (remove the oldest Alpine ice core records).

In the same sentence, please remove "back to the last northern hemisphere climatic optimum": Indeed, as discussed in their review, Heiri et al. (2014), a number of climate records imply a Holocene Climate Optimum in the Alps during the 10 to 5 ky BP period but often with different timing. On the other hand, the discussion of the DDG record (Legrand et al., 2025) revealed that as for some other European records no evidence of a Holocene Climate Optimum is observed in the DDG ice record, opposed to Greenland records that show a well-marked Climate Optimum. Anyway, having mentioned the climatic optimum in the abstract, you do not come back to this point when presenting the $\delta^{18}O$ record (Figure 1) in the manuscript. If this discussion is foreseen as the geochemical discussion of Pb, for another paper, I wonder whether the actual manuscript should rather go to The Cryosphere journal than to Climate of the Past.

Heiri, O. *et al.* Palaeoclimate records 60–8 ka in the Austrian and Swiss Alps and their forelands. *Quaternary Science Reviews* 106, 186-205 (2014). https://doi.org/10.1016/j.quascirev.2014.05.021
Legrand, M., McConnell, J. R., Preunkert, S., Wachs, D., Chellman, N. J., Rehfeld, K., Bergametti, G., Wensman, S. M., Aeschbach, W., Oberthaler, M., & Friedrich, R. (2025), Alpine ice core record of large changes in dust, seasalt, and biogenic aerosol over Europe during deglaciation. PNAS Nexus, Vol. 4 (6), doi: https://doi.org/10.1093/pnasnexus/pgaf186

2) Question and comments concerning the Ortles Pb records:

- Page 3 Line 30: The wording "crustal excess Pb concentration" sounds strange: you mean the non-crustal Pb (ncPb) ?

- An additional section is needed before section 2 (ice core realignment) reporting on the different analytical approaches for Pb measurements applied to Ortles core 1 and 2, and compared to the one applied for the AN core. Also specify in this section how ncPb was calculated (which crustal reference species is used and what is assumed Pb/crust ratio?). Please give blanks and detection limits and introduce a discussion on the effect (or not) of the online acidification (there are numerous papers on the literature on that) (see also next comment).

- Page 4 Lines 22-25 (section 2): the leaching time.
The authors seem to ignore numerous publication that examine the recovery of species using the online acidification of the CFA systems. For instance, previous assessment of measurement recovery during continuous measurements with the DRI system indicated that recovery was 100% and 60% for Pb and Ce, respectively (McConnell et al., 2018). That is because Ce is mainly present in relatively large dust particles and so remains in the particle phase longer than other elements associated with pollution that are adsorbed onto smaller particles and so readily washed off during on-line acidification. Other studies on this topic include Arienzo et al. 2019 (and references therein). Generally, these studies indicated a good recovery for Pb (in contrary to Al or Fe for instance). One of the reasons is that Pb is one of the trace elements that have very weak crustal contribution. For instance, Preunkert et al. (2019) indicated (in their Figure 2) that at the Col du Dome Alpine site the crustal lead contribution is negligible even during the pre-roman antiquity when the Pb level was as low as 0.015 ng g$^{-1}$.

With that in mind, let's have a look on Figure 2 of the manuscript and the comparison of the Pb magnitudes between core 1 and core 3. However before doing so, note that the y axis title is wrong, "pg/g$^{-1}$" makes no sense, and compared with Figure 8 obviously it should not be pg g$^{-1}$ but ng g$^{-1}$.
At 59-60 m depth in core 1 there are ~0.44 ng g$^{-1}$ instead of ~0.13 ng g$^{-1}$ in core 3, at 65-65.5 m depth in core 1 there are ~ 4.5 ng g$^{-1}$ instead of ~ 1.8 ng g$^{-1}$ in core 3, at 68-69 m depth in core 1 there are ~ 2.2 ng g$^{-1}$ instead of ~ 1.0 ng g$^{-1}$ in core 3, and at 69.5-70 m in core 1 there are ~ 0.8 ng g$^{-1}$ instead of ~ 0.02 ng g$^{-1}$ in core 3. These differences are significant with respect to absolute Pb fluctuations over time (of course they are less visible on the log scale shown in Figure 2). Why do this differences between the two cores suddenly disappear at certain depths? If attributed to a problem of leaching (as suggested in the manuscript) one would expect an increase of crustal species at depths for which the differences between the two cores are important. Checking the Rb data provided following a question from a reviewer during the CP2022 review process, I saw however no large enough changes being able to disturb the Pb data gained with the online acidification. From that I conclude that the reason which is invoked to explain the Pb record differences between core 1 and core 3 is not related to leaching time but rather suggests that core 1 might data have suffered from contamination.

McConnell, J. R., Wilson, A. I., Stohl, A., Arienzo, M. M., Chellman, N. J., Eckhardt, S., et al. (2018). Lead pollution recorded in Greenland ice indicates European emissions tracked plagues, wars, and imperial expansion

during antiquity. *Proceedings of the National Academy of Sciences of the United States of America*, *115*(22), 5726–5731. https://doi.org/10.1073/pnas.1721818115

Arienzo, M. M., McConnell, J. R., Chellman, N., & Kipfstuhl, S. (2019). Method for correcting continuous ice-core elemental measurements for under-recovery. *Environmental Science & Technology*, *53*(10), 5887–5894. https://doi.org/10.1021/acs.est.9b00199

3) Questions and comments concerning the comparison between Ortles, AN and CG

- Figure 6: In spite of several recommendations during the CP2022 review process, to provide a better Figure 6, the present version is still unreadable even with the linear scale for AN. It is a pity since this Figure illustrates how you propose to adjust peaks by comparing AN and Ortles. You propose to shift the oldest peak in Ortles from 400 BCE (dated with Ortles [14]C) to 180 BCE to become the roman Republic peak, and the wide peak dated with Ortles [14]C to 350 BCE to 510 CE to 70 BCE to 230 CE, to become the roman Empire perturbation. Thus, between the oldest peak and the end of roman Empire perturbation initially covering 860 years is reduced to 300 years. That is almost a factor of 3 and would require more comments.

- Also, there are other major differences between the Ortles and AN Pb record (see Figure 6 and 8), i.e. from 400 to 700 CE and after 1400 CE to recent times) that strongly require geochemical comments. In addition, the peaks dated with Ortles 14C to 600 CE and 1400 CE are higher than the industrial Pb increase at Ortles. This is in contrary to all published Pb ice core trends from Greenland and the Alps (CG and CDD). Again, a geochemical discussion would be needed here.

- Why do you compare the Pb records between Ortles and the remote site of AN? On the one hand, I agree that it is legitimate given the accuracy of the AN dating, on the other hand you missed to check other relevant records such as the peatbog record from Tyrol (70 km away from your site, von Scheffer et al., 2024) that shows in its Figure 5 a large increase of Pb from 400 BCE to ~500 CE, i.e. quite similar to your record (without the scaling on AN). Again, if you do not want to discuss the geochemistry of Pb, I am not convinced by the reliability of the applied peak matching between Ortles and AN. Also note that, the applied Pearson Correlation is not an adequate measure to assess the goodness of the peak matching, since it is unsensitive against whether in reality same or different dated peaks are matched together.

- Whereas I agree that a comparison between the Roman Pb perturbation shown at the Col du Dome (CDD) site by Preunkert et al. (2019) and recently at DDG (Legrand et al., 20325) is not evident since the published Pb records are not as continuous as at AN, a direct comparison of Pb concentrations observed over the Roman perturbation with other Alpine ice records is more than welcome over the time prior and during the roman antiquity, and more relevant that a comparison of ice concentrations between Ortles and AN.

- In Fig S2 the Ortles Pb record is compared with the Pb CG03 record from CG but unfortunately the roman perturbation is not present in this latter record. What is the reason the comparison with CDD and DDG records have not been conducted?

Von Scheffer, C.; De Vleeschouwer, F.; Le Roux, G.; Unkel, I. Mineral dust and lead deposition from land use and metallurgy in a 4800-year-old peat record from the Central Alps (Tyrol, Austria). *Quater. Inter.* **2024**, *700–701*, 68–79.

4) Conclusion:

Rephrase conclusion, depending whether the Pb record is kept and its alignment to AN is discussed geochemically. If it is kept, add the fact that the very reliable [14]C age assignments based on larch and charred spruce needles, were shifted for more than 1-sigma in the lower part of the core to match the Pb peaks. And rephrase the last sentence of the conclusion in view of the now existing western Alpes CDD record including [14]C, [39]Ar, the Pb roman antiquity perturbation and the drop of $\delta^{18}O$ when entering the end or mid Younger Dryas period.

5) Other comments:

- Different depth units are used within the manuscript: depth in meter with 0 m at the glacier surface, depth in m w.e. with 0 m we at the glacier surface, ice thickness in m w.e. with 0 at glacier bedrock. No conversion in given between the different scales even not from m to m w.e.. With that, the reader cannot compare data between the different figures and Tables. Use only one single depth unit (m w.e. would be probably the most adequate) within the whole manuscript.

- Supplementary Text S2: The first sentence is very misleading: "The Colle Gnifetti ice core (Mt. Rosa, Western Alps) is currently the oldest record from the Alps, dating back >15000 5 years (Jenk et al., 2009)". This sentence gives the impression that the CG ice recorded environments (and/or climate) back to more than 15,000 years. This is however not correct: as argued by Jenk et al. (2009), while radiocarbon analyses of particulate organic carbon have indicated that Pleistocene ice is sometimes present in the bottom layers at CG, it is shown that prior to 3,000 years the climate $\delta^{18}O$ record was strongly disturbed by post-deposition liquid migration of [18]O at the grain boundary of ice located in zones of strong strain-rate gradients above the inclined bedrock.

- The captions of figures are often not well completed. For instance, in Figure 6, 8, Fig S2, where you report the AN record of Pb, you have to specify in the caption that the AN record is from Mc Connell et al. (2019) (the citation only in the text is not enough). Please check also for CG.

- Figure 6, 8, S2 and S3: Records on linear scale are still unreadable, again please change the scales. For example, you could increase the height of the graphs and cut the y axis scale at 3 ng g[-1] for Ortles and CG03 and at 0.3 ng g[-1] for AN (indicating the maximum of the industrialization with a flesh to the top and a number). This would put the 10-year averages in the focus. Anyway, the annual lines indicated are too thin to be visible.

- Fig S2, As for Ortles and AN (see Figure 6 and 8), the discrepancy between CG03 and AN after 1600 CE is huge (even on a log scale). That requires a comment in the text. Also, in Fig S3 the comparison between Ortles and CG03 requires a discussion.

- Tables: in all tables the type sizes are too small

- no data were provided to the reviewer for the review process. Please, detail which data will be made available in the World Data Center. If the Pb data are kept in the manuscript they need to be made available.

---

## Author Comment (AC2)

**Review 2**

- Does the paper address relevant scientific questions within the scope of CP? If geochemically discussion of Pb records will be addressed in a revised version (see comments below), yes, if not, no.
- Does the paper present novel concepts, ideas, tools, or data? No, except for Pollen analysis
- Are substantial conclusions reached? Not really, but already existing ice core dating was improved.
- Are the scientific methods and assumptions valid and clearly outlined? No, see comments below
- Are the results sufficient to support the interpretations and conclusions? No, see comment below
- Is the description of experiments and calculations sufficiently complete and precise to allow their reproduction by fellow scientists (traceability of results)? No, see comments below
- Do the authors give proper credit to related work and clearly indicate their own new/original contribution? Yes, but some references are missed
- Does the title clearly reflect the contents of the paper? No since comparison with Arctic is not geochemically discussed
- Is the overall presentation well structured and clear? Yes, but analytical section is missing
- Is the language fluent and precise? Yes
- Should any parts of the paper (text, formulae, figures, tables) be clarified, reduced, combined, or eliminated? Yes, see comments below

We thank Reviewer 2 for the detailed evaluation of our manuscript and constructive suggestions.

**Overall comment:**

There is a) an absence of analytical details

In the revised version of the manuscript we will add more information regarding the different analytical methods (discrete and CFA measurements), as well as details about blanks and detection limits.

b) a not very serious discussion on the difference between the Pb records of Ortles core 1 and core 3,

Please, see our response to a related comment by Reviewer 1.

and c) an absence of a geochemical reasoning for the validity of the comparison between the Ortles and the AN Pb record.

We will more clearly outline the approach, reasoning and geochemical assumptions that are instrumental to the main focus of this study that aims at improving the chronology of the Mt. Ortles ice cores only. In particular, we will add some more geochemical context and referencing about Roman Pb and Pb mined in the Alps over the presented time scale.

On this last point, on the one hand, the authors argue that the geochemical discussion of the Ortles Pb records are out of the scope of this paper, but on the other hand they adjust two Ortles Pb peaks (dated by Ortles 14C to environ 400 and 230 BC) on the two AN peaks attributed by Mc Connell et al. (2019) to the roman Republic and Empire (70 BC and 100 CE), without clearly specifying in the text that this fit is based on the assumption that Ortles ice has recorded the Roman antiquity, and without discussion on the geochemical reason for this assumption, respectively.

We will certainly clarify and extend this in a revised version of the manuscript (please see below our extensive explanation and technical justification for peak attributions).

This is incomprehensible and incoherent already (see further comments below) because there is a potential alternative interpretation of this two peaks in the Ortles Pb record which leaves them at the age attributed by the 14C and which is not presented in the manuscript.

Importantly, here we note that the ages of the Pb concentration peaks remain, *within the linked uncertainty*, the same ones which were initially attributed by 14C, even after synchronization. The critical point that needs to be considered is that 14C age markers always indicate a time range based on probability. Notably, the shifts we apply are within that probability range. Please see below our detailed explanation and technical justification.

Also, there are major differences between the Ortles and AN Pb records (see Figure 6 and 8), e.g. from 400 to 700 CE and after 1400 CE to recent times, which strongly require geochemical comments.

Here we anticipate that in the revised manuscript we will discuss how local (Tyrolean) and remote (continental) sources of atmospheric Pb for the Mt. Ortles drilling site are likely to modulate the amplitude of Pb concentrations at this high altitude Central European drilling site. In fact, Mt. Ortles is most likely influenced by these two kinds of distinct contributions whereas these same sources will affect with a different intensity a much more distant Arctic drilling site like AN.

Given also numerous errors, the very poor quality of Figures reporting on Pb (some are completely unreadable), the non-convincing explanation for differences of the Pb records in core 1 and 3, and finally the quasi-absence of a plus value of the comparison between Ortles and AN (if correct),

We respectfully disagree with the Reviewer's assessment that we made numerous errors. and that the figures are of very poor quality. However, there is always room for improvement and we remain open to specific suggestions and we will perform some changes (please, see below).Regarding the observed discrepancies in the absolute Pb concentrations between core 1 and 3, please see our reply to Reviewer 1.

As stated in various parts of the manuscript, the plus value of the comparison between the Mt. Ortles and AN Pb records is a remarkably improved chronology for the Mt. Ortles ice cores (both in dating accuracy and reduction in dating uncertainty). In the revised version of the manuscript we will further remark the additional value of synchronizing well-dated polar cores with high altitude-low latitude cores that typically show a much larger age uncertainty. In particular we will state that considering absolute dates constraints is required to prevent arbitrary shifts of the record over time.

I strongly recommend to refocus the paper on the other dating improvements only and skip the whole presentation of Pb.

As stated above and detailed below, we have a different and justified view on this point as the synchronization of two Pb polar and alpine records is an important motivation and element of interest for this manuscript that leads to a more accurate and useful timescale. Indeed, before any environmental or climate record from a natural archive can be discussed and interpretated, it is crucial to first assure the best possible chronology of the

records. As we notice that the main motivation of this study is still unclear, we will further clarify it in the revised version of the manuscript.

If the authors decide to maintain the Pb part of the manuscript, they have to provide a complete geochemical discussion (and revise the text with respect to several problems that will be detailed below).

As mentioned above we plan to significantly expand the geochemical discussion in a way that is functional to the purpose and focus of this study linked to improving a chronology which we revised using a variety of methods and approaches presented aside from the Pb synchronization.

**Specific comments:**

**1) Abstract:**

Whereas at the time of submission, the authors could not have known about recent findings made in the French Alps at the ice core drill site of Dome du Goutier (DDG) showing undisturbed continuous climate and aerosol ice records spanning the last 12,000 years (Legrand et al., 2025), however this new finding should be referenced and the wording in the text adequately updated.

We thank Reviewer 2 for pointing out the new manuscript that was published after our submission. We will consider and reference it in the revised version of our manuscript.

For instance, the first sentence of the abstract (line 20) is not true anymore: please change to "provided evidence that the record spans the last 7,000 years" (remove the oldest Alpine ice core records).

We note that in this sentence from the abstract we refer to the Ortles ice core just as "one of the oldest ice core records". Thus this sentence remains correct and does not need to be udated.

In the same sentence, please remove "back to the last northern hemisphere climatic optimum": Indeed, as discussed in their review, Heiri et al. (2014), a number of climate records imply a Holocene Climate Optimum in the Alps during the 10 to 5 ky BP period but often with different timing. On the other hand, the discussion of the DDG record (Legrand

et al., 2025) revealed that as for some other European records no evidence of a Holocene Climate Optimum is observed in the DDG ice record, opposed to Greenland records that show a well-marked Climate Optimum.

As the Reviewer acknowledges, there is a number of studies which discuss a Holocene Climate Optimum (HCO) in the European Alps, among them the mentioned Heiri et al. (2014), but also a large number of studies with robust evidence of significantly reduced glacier extend during that period (here we just cite two out of the many: e.g., Joerin et al, 2007; Kutschera et al., 2020). There might be different causes (e.g., shift in seasonal distribution of precipitation and/or accumulation), but we do not think our manuscript (or the response here) is the appropriate venue to discuss why no evidence of the HCO was found in the DDG record.

Also, this HCO related sentence in the abstract, was just a reference to our first publication of the Mt. Ortles ice cores dating "Age of the Mt. Ortles ice cores, the Tyrolean Iceman and glaciation of the highest summit of South Tyrol since the Northern Hemisphere Climatic Optimum" (Gabrielli et al., 2016). In any event, we are not planning to discuss further the age of the Mt. Ortles bottom ice in this manuscript as this portion of the chronology has not been revised and remains essentially identical to the one already published in 2016.

Anyway, having mentioned the climatic optimum in the abstract, you do not come back to this point when presenting the d18O record (Figure 1) in the manuscript. If this discussion is foreseen as the geochemical discussion of Pb, for another paper, I wonder whether the actual manuscript should rather go to The Cryosphere journal than to Climate of the Past.

We do believe this manuscript can be of the highest interest for the readers of Climate of the Past as it presents a revised and exceptionally accurate alpine chronology that can attract the attention of a broader audience involved in the studies of a variety of multiple paleoclimate archives.

**2) Question and comments concerning the Ortles Pb records:**

- Page 3 Line 30: The wording "crustal excess Pb concentration" sounds strange: you mean the non-crustal Pb (ncPb) ?

Thank you for the suggestion. We fully agree and will change the wording accordingly throughout the manuscript.

- An additional section is needed before section 2 (ice core realignment) reporting on the different analytical approaches for Pb measurements applied to Ortles core 1 and 2, and compared to the one applied for the AN core.

We will move and expand the current discussion on acid leaching in the Supplementary Information.

Also specify in this section how ncPb was calculated (which crustal reference species is used and what is assumed Pb/crust ratio?).

As already stated in the manuscript and in the Supplementary Information:

"*The non-crustal Pb concentration was obtained by using the Pb/Rb ratio of 0.51 (value obtained from a deep core #3 section characterized by the lowest Pb concentrations between 69.55 and 69.94 m depth)*".

"*For the Colle Gnifetti core, the non-crustal Pb concentration record was obtained by using the average crustal Pb/Ti ratio (0.00545 obtained from Wedepohl, 1995)*".

Please give blanks and detection limits and introduce a discussion on the effect (or not) of the online acidification (there are numerous papers on the literature on that) (see also next comment).

Please, see our comments to related request above. Here we remark once again that while differential acid leaching may alter absolute Pb concentration values and perhaps the size of amplitude between maxima and minima, different acid leaching procedures do not impact the variations/fluctuations in the record (shape) and thus have no practical implications on the Pb alignments/match.

- Page 4 Lines 22-25 (section 2): the leaching time. The authors seem to ignore numerous publication that examine the recovery of species using the online acidification of the CFA systems. For instance, previous assessment of measurement recovery during continuous measurements with the DRI system indicated that recovery was 100% and 60% for Pb and Ce, respectively (McConnell et al., 2018). That is because Ce is mainly present in relatively large dust particles and so remains in the particle phase longer than other elements associated with pollution that are adsorbed onto smaller particles and so readily washed off during on-line acidification. Other studies on this topic include Arienzo et al. 2019 (and references therein). Generally, these studies indicated a good recovery for Pb (in contrary to

Al or Fe for instance). One of the reasons is that Pb is one of the trace elements that have very weak crustal contribution. For instance, Preunkert et al. (2019) indicated (in their Figure 2) that at the Col du Dome Alpine site the crustal lead contribution is negligible even during the pre-roman antiquity when the Pb level was as low as 0.015 ng g-1

We are fully aware of this topic (some of us were even co-authors on some of such studies), but we accidentally removed the Arienzo et al. 2019 reference which was present in our previous response during the 2022 submission to CP. We certainly will add it again in the revised manuscript, as well as some additional references (see response to a related comment above). Here we just remark that Pb leaching is site depend on the local dust characteristics and mineralogy and cannot be generalized.

With that in mind, let's have a look on Figure 2 of the manuscript and the comparison of the Pb magnitudes between core 1 and core 3. However before doing so, note that the y axis title is wrong, "pg/g-1" makes no sense, and compared with Figure 8 obviously it should not be pg g-1 but ng g-1

Good catch, thank you very much. We will correct this.

At 59-60 m depth in core 1 there are ~0.44 ng g-1 instead of ~0.13 ng g-1 in core 3, at 65-65.5 m depth in core 1 there are ~ 4.5 ng g-1 instead of ~ 1.8 ng g-1 in core 3, at 68-69 m depth in core 1 there are ~ 2.2 ng g-1 instead of ~ 1.0 ng g-1 in core 3, and at 69.5-70 m in core 1 there are ~ 0.8 ng g-1 instead of ~ 0.02 ng g-1 in core 3. These differences are significant with respect to absolute Pb fluctuations over time (of course they are less visible on the log scale shown in Figure 2). Why do this differences between the two cores suddenly disappear at certain depths?

As mentioned, differences in the absolute concentrations of Pb fluctuations, cannot affect the conclusions presented in this paper. In any case, it is possible that differences might be larger at low Pb concentrations where the Pb leaching from mineral dust particles of relatively large size can become relatively more important even at low crustal dust concentrations.

If attributed to a problem of leaching (as suggested in the manuscript) one would expect an increase of crustal species at depths for which the differences between the two cores are important.

Not necessarily. It may also depend on the dust size / surface with a larger surface/ leaching at the smallest dust size.

Checking the Rb data provided following a question from a reviewer during the CP2022 review process, I saw however no large enough changes being able to disturb the Pb data gained with the online acidification.

Low Pb levels are more likely disturbed by Pb mineral leaching even at very low crustal species concentration level. In addition, crustal species leaching Pb are not necessarily the same bearing – Rb mineral species. Thus Pb mineral leaching and Rb concentration levels are not necessarily correlated.

From that I conclude that the reason which is invoked to explain the Pb record differences between core 1 and core 3 is not related to leaching time but rather suggests that core 1 might data have suffered from contamination.

This is unlikely as no indication of significant Pb contamination was found, based on the procedural blanks. While we think the acid leaching topic will be of interest in a future manuscript focused on past Pb sources and emissions, the acid-leaching process is of minor relevance in the context of this manuscript where only the shape of the record can affect the conclusions (revised chronology).

**3) Questions and comments concerning the comparison between Ortles, AN and CG**

- Figure 6: In spite of several recommendations during the CP2022 review process, to provide a better Figure 6, the present version is still unreadable even with the linear scale for AN. It is a pity since this Figure illustrates how you propose to adjust peaks by comparing AN and Ortles.

We respectfully disagree as we made major changes to Figure 6 when compared to the previous version (CP2022 revised version). Concentration levels are now shown both, on the linear and on the log scale. In the panel with data plotted on the linear scale, the peaks which were assigned are now immediately visible and can directly be compared to each other in terms of absolute concentrations. It is correct that small scale variations and the minima are hardly visible but the current panel showing the data on the log scale accounts for that. In this panel, the small-scale fluctuations can easily be observed while they also can be transferred to absolute values (linear scale). The panel on the log scale has a

second purpose as it allows for a facilitated visible recognition of the characteristic fluctuations over time, for both, minima, and maxima which, for the purpose of visualize the alignment, is most relevant. We conclude that the changes made are major and fulfill the requirements while still portraying what is most important for this study. Nevertheless, if an increase in the font size is the issue and is required by the journal, we can certainly make this change.

You propose to shift the oldest peak in Ortles from 400 BCE (dated with Ortles 14C) to 180 BCE to become the roman Republic peak, and the wide peak dated with Ortles 14C to 350 BCE to 510 CE to 70 BCE to 230 CE, to become the roman Empire perturbation. Thus, between the oldest peak and the end of roman Empire perturbation initially covering 860 years is reduced to 300 years. That is almost a factor of 3 and would require more comments. Also, there are other major differences between the Ortles and AN Pb record (see Figure 6 and 8), i.e. from 400 to 700 CE and after 1400 CE to recent times) that strongly require geochemical comments.

Uncertainties of the absolute time markers used for dating are very important and should always be considered. In this context, it is important to note that we shift major peaks within boundaries constrained by the age probability ranges (e.g. based on the absolute dates from 14C) which are consequently propagated to any depth through the applied COPRA model).

More to the point, before applying synchronization of CP2025/1 with AN (in the manuscript CP2025/1 denotes the intermediate timescale), the "oldest" peak in Pb concentration in the Mt. Ortles cores (the peak maxima; indicated with 1/yellow in the Figure 1 presented below) is assigned to an age between 510-320 BCE (intermediate age in CP2025/1). After synchronization, that peak maxima is shifted to 205-145 BCE (final age in CP2025/2). Please note that these age ranges indicate the 1σ probability. On page 12, line 34 in the manuscript we wrote: "Importantly, 7 of the 11 tie points were within the 1-sigma uncertainty of the initial CP2025/1 timescale built based on the absolute ages from 14C, with the other 4 being within the 2-sigma range...". The discussed point was indeed one (out of 4) where the shift from the intermediate scale (CP2025/1) was consistent within the 2-sigma boundary. However, it should also be noted that, when compared to the previously published timescale TC2016 (Gabrielli et al., 2016), the newly proposed final age for that depth is older by 20 years only.

The other, younger peak pointed out by the Reviewer 2, is indicated with 2/yellow (see figure below). The intermediate scale CP2025/1 assigned it to an age between 350+/-110 BCE and

515+/-270 CE. The covered age span derived after synchronization is between 65+/-30 BCE and 245+/-50 CE. Regarding the reduction of covered timespan between the two peaks (1/yellow and 2/yellow), as obtained by the intermediate dating in CP2025/1 and the final age scale CP2025/2, the Reviewer is not entirely correct in its calculation as the time compression is not from 860 to 300 years, but in fact from 500 (CP2025/1: peak 1 – peak 2) to 265. This is about a factor of 2. If we also consider dating uncertainties, the timespan according to CP2025/1 can be between 220 and 780 years, and according to CP2025/2 between 190 and 340 years. This shows, that based on the uncertainty of our absolute ages from 14C, both peak shifts operated by CP2025/1 and CP2025/2 are possible. Our hypothesis is that the correct shift is the one displayed by CP2025/2, which, importantly, is additionally constrained based on the overal shape of the Pb signal (see Figure 1 below). The underlying assumption (to be more extensively presented within the revised manuscript) is that a Pb signal recorded in the Arctic with an emission source located much closer to Mt. Ortles, should be imprinted also in the Mt. Ortles ice at approximately the same time, thus displaying a distinct signal also in the Mt. Ortles Pb ice core record.

[Figure]

*Figure 1: Panels a and b refer to the relevant time sections and are cropped from our current Fig. 6. Panel c is an adaptation of our Fig. 8 for this discussion. Panel d is the same as panel c, but with the normalized non-background lead (nbPb) flux signal from a composite of three Arctic ice cores (McConnell et al., 2025, Fig. 1 therein). All data are displayed as a 10-year average to be consistent with the smoothing applied to the other datasets presented. Please see within the text of this response for a description of numbers highlighted with colors.*

The presumed "wide peak" pointed out by the Reviewer is a good example for an explanation. When plotted on the initially derived intermediate age scale CP2025/1, it has no well-defined (clear and single) peak maxima (panel a and b). In this case, it is impossible to assign any Mt. Ortles Pb concentration peak to the correct (i.e., an associated) peak observed in AN. Thus, using that signal for synchronization would be highly questionable and therefore we did not consider it. In contrast, we considered the minima 3/green and 4/green as indicated in the figures. As a consequence, the final age of the maxima and the final age span of peak 2 ( "wide peak") was defined without assigning a specific date to the peak 2/yellow.

Looking at the final temporal shape/structure of the Mt. Ortles Pb concentration signal over the Roman period from 200 BCE to 300 CE and comparing it to the signal recorded (observed) in the Arctic record from AN (panel c in the above Figure), this now shows a stunning similarity, even though there was no additional matching during that period. In addition, here (panel d) we compare it as well to the Arctic 3-ice core Pb composite from McConnell et al., 2025 (in which AN is one of three records used). Also in this case the signal from 500 BCE to 300 CE appears very highly correlated considering that we only shifted our ages within the set boundaries (2sigma, and 1sigma in most cases).

Despite the underlying idea and assumptions of the above-described approach are already contained in the submitted manuscript, a short paragraph, summarizing this (and potentially more; see below) will be added in in the Supplementary Information in a revised version.

In addition, the peaks dated with Ortles 14C to 600 CE and 1400 CE are higher than the industrial Pb increase at Ortles. This is in contrary to all published Pb ice core trends from Greenland and the Alps (CG and CDD). Again, a geochemical discussion would be needed here.

Our current hypothesis is that old/local sources with high emission factors vs. modern/more distant sources might partly explain the observed Pb trend in the Mt. Ortles ice core. It is also important to note, that the Mt. Ortles glacier archive at the study site is temperate in the upper firn part only (down to 30 m depth; see Gabrielli et al 2012) and post depositional effects, causing a partial loss of the most recent Pb signal by melt-water runoff are possible (see e.g., Huber et al., 2024; and Avak et al., 2018, estimating up to 50% loss for the Pb signal). In any event, the recent modern period is dated by annual layer counting using more conservative parameters to melt water percolation such as pollen, while Pb synchronization is not being applied during this period. Thus this comment will be

extensively discussed in future publications. Neverthelss, we will briefly anticipate this explanation in the revised version of this manuscript.

- Why do you compare the Pb records between Ortles and the remote site of AN?

The explanation is already provided in the manuscript (page 12, line 10-19): *"In general, well-dated Pb records from six Arctic cores (McConnell et al., 2019) reaching back to 200 BCE show similarities with the high-resolution Pb record from the Alto dell'Ortles ice core (Fig. 6). Among the Arctic Pb records, the one from the AN ice cap in Severnaya Zemlya, Russian Arctic was selected for synchronization with the Alto dell'Ortles cores (Fig. 6) because: i) it contains one of the longest continuous chronologies; ii) it provides the largest Pb amplitude signal; and, iii) it is suggested by atmospheric modelling to be the most influenced by Pb emissions from silver mining and metallurgical activities in central Europe close to Mt. Ortles during the past millennia (McConnell et al., 2019), making it more likely that the Alto dell'Ortles and AN drilling sites share similar variability in atmospheric Pb deposition. The uncertainty of the AN chronology was estimated to be less than 5 years between 500 and 1999 CE, with less than 2 years uncertainty at the known volcanic time horizons within the Greenland reference record (McConnell et al., 2019)."*

In a revised version, we will further clarify this by adding a geochemical justification of our working hypothesis and reasoning.

On the one hand, I agree that it is legitimate given the accuracy of the AN dating, on the other hand you missed to check other relevant records such as the peatbog record from Tyrol (70 km away from your site, von Scheffer et al., 2024) that shows in its Figure 5 a large increase of Pb from 400 BCE to ~500 CE, i.e. quite similar to your record (without the scaling on AN).

We note that the "quite similar" statement by the Reviewer is rather subjective and qualitative. We checked it carefully and we cannot confirm it (see Figure 2 below). The dating of the peatbog record from Scheffer et al., 2024 is based on six 14C dates between around 0 BP to 5000 BP. Based on its Fig. 2, the uncertainty of the dating for the period 400 BCE to ~500 CE can be estimated to be between 100 to 200 years. This is similar to the uncertainty of our initially published TC2016 timescale, and therefore it cannot be helpful to obtain an improved accuracy, nor a better precision.

[Figure]

*Figure 2: The panels on the right and on the left refer to comparisons of the Mt. Ortles Pb concentration record with the Scheffer et al. 2024 record, before and after the synchronization of the Mt. Ortles Pb concentration with the AN Arcticrecord, respectively.*

Again, if you do not want to discuss the geochemistry of Pb, I am not convinced by the reliability of the applied peak matching between Ortles and AN.

Our procedure is based on robust statistics and probabilities. Age markers probabilities are further constrained by the assumption that a Pb signal recorded in the remote Arctic should also be present in a record from a glacial archive (Mt. Ortles) which is much closer to the known few major emission sources existing in the antiquity in the Northern Hemisphere. This will be discussed in the revised version of our manuscript.

We propose this approach that we hope can  initiate a more general discussion within a larger science community. The bottom line is that, while wiggle matching without constraint from absolute dates has been done frequently in many other past studies, here we show that boundaries from absolute age markers must initially exist and be taken into account, for the wiggle matching to be justified and useful. We will also remark this important point in the revised version of this manuscript.

Also note that, the applied Pearson Correlation is not an adequate measure to assess the goodness of the peak matching, since it is unsensitive against whether in reality same or different dated peaks are matched together.

The Pearson-Correlation is used by the AnalyseSerie software as one of the metrics to evaluate the wiggle matching of the record. Another metric used is the accumulation rate that must remain reasonable in order to (un)-compress realistically the records. As mentioned, other constrains to wiggle matching are the boundaries from the absolute age markers. Once more, in our paper we do not simply and arbitrarily match different peaks and we do believe this is a major methodological advance in dating high altitude – low latitude ice cores.

- Whereas I agree that a comparison between the Roman Pb perturbation shown at the Col du Dome (CDD) site by Preunkert et al. (2019) and recently at DDG (Legrand et al., 20325) is not evident since the published Pb records are not as continuous as at AN, a direct comparison of Pb concentrations observed over the Roman perturbation with other Alpine ice records is more than welcome over the time prior and during the roman antiquity, and more relevant that acomparison of ice concentrations between Ortles and AN.

Certainly, this is of interest and as mentioned in the manuscript, this is planned in a future publication. while we believe it is out of scope in the current manuscript.

- In Fig S2 the Ortles Pb record is compared with the Pb CG03 record from CG but unfortunately the roman perturbation is not present in this latter record. What is the reason the comparison with CDD and DDG records have not been conducted?

There are two reasons: 1) the DDG was not published by the time of our submission; 2) the uncertainty of dating. In fact the CG record has an annual counted dating back to 1763 CE. As a consequence, the observed similarity of the CG Pb signal with the Arctic Pb record extends back to 1500 CE. Before 1500 CE the CG dating uncertainty increases rapidly. Here we notice that the point 2 presented above is an important piece of information in the overall context of the manuscript and will be reported more clearly in the revised version of our manuscript.

**4) Conclusion:**

Rephrase conclusion, depending whether the Pb record is kept and its alignment to AN is discussed geochemically. If it is kept, add the fact that the very reliable 14C age assignments based on larch and charred spruce needles, were shifted for more than 1-sigma in the lower part of the core to match the Pb peaks.

For the various reasons outlined in the responses presented above, we definitely intend to keep the Pb synchronization in our manuscript, but, as mentioned above, we will also clarify the basic, underlying the probabilistic and geochemical assumptions of our approach.

We will also add a statement regarding the spruce needle, and in particular we will also mention that a) the 14C age of the larch leaf was not shifted in this revised timescale, b) the revised chronology was shifted from the initial chronology TC2016 by a few decades only,

and is well within the uncertainties; and c) that individual 14C data points are not as accurate as the Reviewer 2 seems to assume (even 14C dates on macrofossils can scatter by several hundred years; e.g., see Thompson et al., 2002 and SI thereof).

And rephrase the last sentence of the conclusion in view of the now existing western Alpes CDD record including 14C, 39Ar, the Pb roman antiquity perturbation and the drop of d18O when entering the end or mid Younger Dryas period.

The Reviewer thinks this sentence should be changed (*"In conclusion, this revised more accurate time scale (CP2025/2) will allow to provide new detailed climatic environmental histories of Central Europe during the Holocene. In addition, CP2025/2 has the potential to become a reference chronology for multiple paleoclimate archives in Europe."*). We agree the sentence should be changed to *"...will allow to provide additional detailed climatic environmental histories of Central Europe during the Holocene..."* (replacing *new* with *additional*).

**5) Other comments:**

- Different depth units are used within the manuscript: depth in meter with 0 m at the glacier surface, depth in m w.e. with 0 m we at the glacier surface, ice thickness in m w.e. with 0 at glacier bedrock. No conversion in given between the different scales even not from m to m w.e.. With that, the reader cannot compare data between the different figures and Tables. Use only one single depth unit (m w.e. would be probably the most adequate) within the whole manuscript.

The following sentence should clarify this point (it is already provided in the SI): *"The DJ model accounts for the importance of density by using m w.e. as unit of length"*. Otherwise, the different depths units are functional to the construction of the time scale and modelling. We will provide specific details about the differences in the main manuscript (i.e., formula how ice thickness in m w.e. translates to depth in m w.e. ( $z_b=H-z_s$, with $z_s$ denoting depth measured from the surface, $z_b$ the depth measured from the bedrock, and $H$ the total ice thickness).

- Supplementary Text S2: The first sentence is very misleading: "The Colle Gnifetti ice core (Mt. Rosa, Western Alps) is currently the oldest record from the Alps, dating back >15000 years (Jenk et al., 2009)". This sentence gives the impression that the CG ice recorded environments (and/or climate) back to more than 15,000 years. This is however not correct:

as argued by Jenk et al. (2009), while radiocarbon analyses of particulate organic carbon have indicated that Pleistocene ice is sometimes present in the bottom layers at CG, it is shown that prior to 3,000 years the climate d18O record was strongly disturbed by post-deposition liquid migration of 18O at the grain boundary of ice located in zones of strong strain-rate gradients above the inclined bedrock.

We will replace the word "*record*" with "*archive*".

- The captions of figures are often not well completed. For instance, in Figure 6, 8, Fig S2, where you report the AN record of Pb, you have to specify in the caption that the AN record is from Mc Connell et al. (2019) (the citation only in the text is not enough). Please check also for CG.

We will include the citations also to the figure captions, as suggested.

- Figure 6, 8, S2 and S3: Records on linear scale are still unreadable, again please change the scales. For example, you could increase the height of the graphs and cut the y axis scale at 3 ng g-1 for Ortles and CG03 and at 0.3 ng g-1 for AN (indicating the maximum of the industrialization with a flesh to the top and a number). This would put the 10-year averages in the focus. Anyway, the annual lines indicated are too thin to be visible.

The records on the linear scale are solely there (per request by the previous review) to clearly highlight the matched peaks. Absolute values for all the data can easily be deduced from the log-scale plots). Since, as already explained above, the main relevance for this study are the "big-scale" variations (not about exact and absolute Pb concentration values), we think making these figures even more complex by introducing an axis break does neither serve the reader nor the clarity and focus of the manuscript.

Regarding the annual lines: considering the dating uncertainty of several decades even after synchronization, here the annual lines are displayed only for completeness (as annual accuracy and precision cannot be achieved by the dating beyond a certain age, as for many other high-altitude ice cores) and to provide some insights about the annual variability. In any case, we will make the annual lines slightly darker.

- Fig S2, As for Ortles and AN (see Figure 6 and 8), the discrepancy between CG03 and AN after 1600 CE is huge (even on a log scale). That requires a comment in the text. Also, in Fig S3 the comparison between Ortles and CG03 requires a discussion.

While this discussion is out of the scope of this paper, we think that a factor 10 difference in absolute Pb concentrations is reasonable, considering different snow accumulation rates, distance, and transport path from the emission sources to the respective sites. Also, please see above the response related to CG .

In a future publication we will discuss, the difference in absolute Pb concentrations while in the revised version of this manuscript we will add a statement about the good agreement between the well-dated (annually counted) record of CG03 back to 1763 CE and the Ortles record after its Pb based age synchronization with AN. This is another confirmation of the validity of the chosen approach.

- Tables: in all tables the type sizes are too small

We fully agree. The problem is fitting large Tables in small A4 pages. This issue will be discussed later with the publisher when dealing with the final format.

- no data were provided to the reviewer for the review process. Please, detail which data will be made available in the World Data Center. If the Pb data are kept in the manuscript they need to be made available.

Pb concentrations of the matched peaks/minima and their linked age prior and after synchronization, will be provided to the public archive. The same applies for the relevant depth/ages markers of d18O and pollen records, which were used here for dating purposes only. All the other data from these records will certainly be provided when these will be discussed in the planned future publications.
* * *
**Additional references:**

Joerin, U. E., et al. (2008). "Holocene optimum events inferred from subglacial sediments at Tschierva Glacier, Eastern Swiss Alps." Quaternary Science Reviews **27**(3): 337-350. https://doi.org/10.1016/j.quascirev.2007.10.016

Kutschera, W., et al. (2020). "The movements of Alpine glaciers throughout the last 10,000 years as sensitive proxies of temperature and climate changes." EPJ Web Conf. 232: 02002. https://doi.org/10.1051/epjconf/202023202002

Huber, C.J., et al. (2024). "High-altitude glacier archives lost due to climate change-related melting." Nat. Geosci. 17, 110–113. https://doi.org/10.1038/s41561-023-01366-1

Avak, S., et al. (2018). "Impact and implications of meltwater percolation on trace element records observed in a high-Alpine ice core." Journal of Glaciology 64(248): 877-886. doi:10.1017/jog.2018.74

Thompson, L.G., et al. (2002). "Kilimanjaro Ice Core Records: Evidence of Holocene Climate Change in Tropical Africa." Science 298, 589-593. doi:10.1126/science.1073198